# Does Progress On Object Recognition Benchmarks Improve Performance on Crowdsourced, Global Data?

**Megan Richards[1], Polina Kirichenko[1,2], Diane Bouchacourt[1], & Mark Ibrahim[1]**
Meta AI[1], New York University[2]
{meganrichards,polkirichenko,dianeb,marksibrahim}@meta.com

## Abstract

For more than a decade, researchers have measured progress in object recognition on the ImageNet dataset along with its associated generalization benchmarks such as ImageNet-A, -C, and -R. Recent advances in foundation models, trained on orders of magnitude more data, have begun to saturate performance on these benchmarks. Despite this progress, even today's best models are brittle in practice. As a step toward more holistic measurement of model reliability, we propose studying performance on crowdsourced, global datasets, which contain natural distribution shifts seen practically in deployment. We perform a comprehensive empirical study on two crowdsourced, globally representative datasets, evaluating nearly 100 vision models to uncover several concerning empirical trends: first, that progress on crowdsourced, global data has significantly lagged behind standard benchmarks, with advances on ImageNet occurring at 2.5x the rate of progress on crowdsourced, global data. Second, we find that progress on standard benchmarks has failed to improve or exacerbated geographic disparities: *geographic disparities between the least performant models and today's best models have more than tripled*. We showcase the promise of using more curated and/or representative training datasets for mitigating these trends, and emphasize curation of web-scale, geographically representative training datasets as a critical open problem for the research community.

## 1 Introduction

ImageNet (Russakovsky et al., 2015), the standard benchmark for object recognition, has set the bar for progress in computer vision. Since its release in 2010, ImageNet along with other generalization benchmarks such as ImageNet-A,-C and -R (Hendrycks et al., 2021b; Hendrycks and Dietterich, 2019; Hendrycks et al., 2021a) has spurred numerous advances in deep learning. Now, more than a decade later, advances in scaling and multi-modal modeling have saturated these standard benchmarks. Most prominently, large-scale vision-language models such as CLIP have been shown to achieve high accuracies on in- and out-of-distribution generalization benchmarks (Radford et al., 2021; Fang et al., 2022a; Miller et al., 2021).

Despite high performance on these benchmarks, model generalization remains an open problem — both vision and text models, as well as state-of-the-art (SOTA) multimodal models, have been found to lack robustness outside of standard benchmarks, even under natural distribution shifts. For example, recent works have shown how CLIP (Radford et al., 2021) remains vulnerable to changes in pose, background, size, position, and lighting (Ibrahim et al., 2022; Madan et al., 2021; Abbas and Deny, 2022; Li et al., 2023a). These results highlight the limitations of commonly used ImageNet generalization benchmarks (also referred to as Out-of-Distribution, or OOD, benchmarks), which focus on controlled, predefined or synthetic alterations of images and do not reflect the rich diversity of data observed during model's deployment (Hendrycks and Dietterich, 2019; Hendrycks et al., 2021b; Madan et al., 2020). We summarize commonly used generalization benchmarks in Table 1.

As a step toward more holistic measurement of model reliability, we propose studying performance on crowdsourced, globally representative datasets. We argue that such datasets offer two distinct

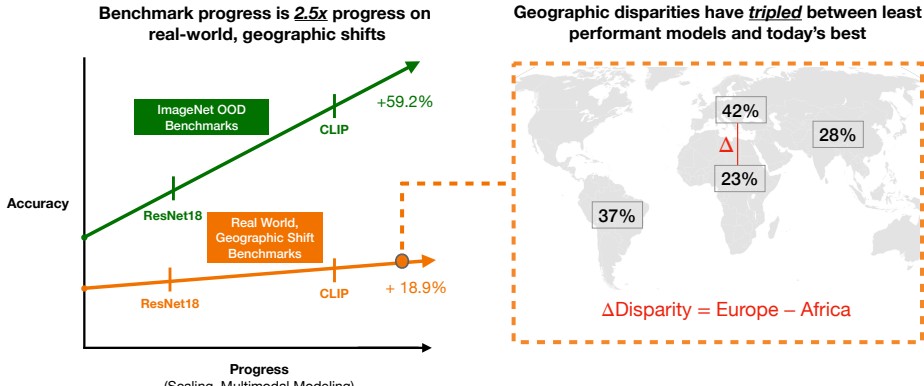

Figure 1: **Progress rate on ImageNet generalization benchmarks is over 2.5x the progress rate on crowdsourced, global data (Section 4)**. Further, geographic disparity between regions is exacerbated with progress on standard benchmarks, tripling over our range of models (Section 5).

advantages missing from current benchmarks. First, they allow us to assess models' performance under *naturally occurring distribution shifts without simulated environments, preselected variations, or artificially injected transformations*. Second, they enable the measurement of geographic disparities between regions, a measure of generalization that is, by definition, relevant across the world, and a critical component of model safety in deployment which is often hidden in the commonly reported average accuracy.

Using these datasets, we offer an extensive empirical analysis of progress, evaluating nearly 100 vision models spanning 16 architectures, 8 distinct pretraining datasets, and a comprehensive set of foundation models. In addition, we systematically study of the impact of common robustness interventions as well as scaling of both model size and data. Our contributions are:

- As a step toward more holistic measurement of model reliability, we propose to measure performance on globally crowdsourced datasets (Section 3), which contain natural distribution shifts relevant to model deployment.
- We identify a significant progress gap, finding progress on ImageNet results in up to 2.5x progress on standard benchmarks than on crowdsourced, global data (Section 4). We illustrate this in the left part of Figure 1.
- We find, contrary to conventional wisdom, that improvements on standard benchmarks exacerbate performance disparities across geographies: disparities in performance have *tripled* between early models and today's best models (Section 5) as shown in the right part of Figure 1.
- We study the impact of common robustness interventions and scaling, finding these two directions are not sufficient to close the progress gap. We explore curating more representative datasets as a promising path to mitigating the trade-offs we uncover (Section 6).

We hope our work will inspire researchers to look beyond standard benchmarks to measure model reliability. To support these efforts, we will release our model test bed and evaluation code in a ready-to-use package, allowing researchers to run their own evaluations with just 4 lines of code.

## 2 RELATED WORK

**Generalization benchmarks have significant limitations.** Model generalization is a major challenge in deep learning. Consequently, a myriad of benchmarks were proposed to evaluate generalization capabilities of image classification models (Recht et al., 2019). For example, ImageNet-A (Hendrycks et al., 2021b) was collected by intentionally mining challenging examples that fool a preselected model. A complementary approach involves applying corruptions to images such as blurring, noise, or style alterations (Hendrycks and Dietterich, 2019; Geirhos et al., 2018a). Other benchmarks such as ImageNet-9 (Xiao et al., 2020), ImageNet-R (Hendrycks et al., 2021a), ImageNet-S (Wang

| Benchmark | shift type | is natural | # shift types | crowdsourced |
|---|---|---|---|---|
| ImageNet | - | - | - | ✗ |
| ImageNet-V2 | - | - | - | ✗ |
| ImageNet-Sketch | drawing | ✓ | 1 | ✗ |
| ImageNet-Rendition | drawing | ✓ | 1 | ✗ |
| ObjectNet | pose, background | ✓ | 3 | ✗ |
| ImageNet-C | corruptions | ✗ | 5 | ✗ |
| ImageNet-A | adversarial | ✓ | 1 | ✗ |
| **DollarStreet** | geographic | ✓ | unlimited | ✓ |
| **GeoDE** | geographic | ✓ | unlimited | ✓ |

Table 1: **Crowdsourced, global data benchmarks enable measuring performance on naturally occurring distribution shifts without simulated environments, preselected variations, or artificially injected transformations**.

et al., 2019) and ObjectNet (Barbu et al., 2019) consist of images with a few predefined axes of generalization in mind: e.g. sketches in ImageNet-S or background variations in ImageNet-9. While being useful measures of model performance under dataset shifts, such benchmarks carry signficant limitations in the kinds of shifts they represent, relying on artificially induced transformations or a predefined criteria, which provide limited measures of object variation in natural settings (DeGrave et al., 2021; Taori et al., 2020).

**Foundation models and robustness interventions.** Many advances from robustness interventions to learning methods leveraging large scale data were proposed to improve generalization. Some robustness interventions are tailored to improve specific generalization axes such as to corruptions (Hendrycks and Dietterich, 2019), texture (Geirhos et al., 2018a), or background shift (Ryali et al., 2021). Data augmentation is a widely used technique which improves generalization (Pinto et al., 2023; Hendrycks et al., 2019; Yun et al., 2019; Li et al., 2023b). Recent work finds that while robustness interventions improve generalization to the intended shift, they may degrade performance to other shifts (Geirhos et al., 2020; Kamath et al., 2021; Moayeri et al., 2022). In parallel, self-supervised models (Goyal et al., 2022; Shi et al., 2022) and more recent foundation models (Pan et al., 2022) trained on much larger datasets (400M text-image pairs) show significant improvements on standard generalization benchmarks (Bommasani et al., 2022). However, in controlled synthetic settings, even large-scale foundation models were found to struggle with common variations in pose, background, and scale, among others (Abbas and Deny, 2022; Ibrahim et al., 2022; Madan et al., 2023). These results highlight that out-of-distribution generalization still remains an open challenge.

**The role of geography in classification.** Geography presents an important axis for measuring model generalization, leveraging a natural distribution shift to measure the consistency of model performance. In recent years, several classification datasets containing images from diverse geographic regions have been developed to study the role of geography in object classification models (Gustafson et al., 2023; Goyal et al., 2021; 2022; Rojas et al.; Ramaswamy et al., 2023).Particularly, Ramaswamy et al. (2023) perform a last-layer retraining experiment with a ResNet50 model and a mixture of ImageNet and GeoDE data, reporting improved performance on GeoDE and DollarStreet datasets. We perform a similar experiment in Sec 6.3 on a ViT model, with only GeoDE data, in order to evaluate last layer retraining as a method to improve region performance disparities. Their analysis reveals that classification models perform much better on some regions compared to others: accuracy gaps across regions can be as high as 20% (DeVries et al., 2019). In conjunction, Shankar et al. (2017a); Dulhanty and Wong (2019); Birhane and Prabhu (2021); Shankar et al. (2017b) present a possible explanation for this performance difference emphasizing over-representation of training images originating from Western geographies. Akin to Dubey et al. (2021) Kalluri et al. (2023) Yin et al. (2023), and Prabhu et al. (2022) which formulates geography as a benchmark for domain adaption, our work presents classification performance gaps across geographic regions as a window into generalization progress.

**Does better in-distribution performance lead to better out-of-distribution generalization?** Chan et al. (2022) shows that generalization in transformer models stems from aspects of the training distribution such as the number and rarity of training classes. Specifically for foundation models such as CLIP, Fang et al. (2022b); Nguyen et al. (2023) show that the main factor driving improved

generalization is the training data quality and distribution (Shi et al., 2023). Miller et al. (2021); Baek et al. (2022) explicitly describe the relationship between in-distribution (ID) and out-of-distribution (OOD) performance, showing ID performance is linearly correlated with OOD generalization. Other work casts doubt on how well ID performance can predict natural OOD generalization (Recht et al., 2019; Teney et al., 2022). Abnar et al. (2021) show improved ID accuracy does not necessarily lead to downstream improvements. Fang et al. (2023) show improvements on ID ImageNet classification does not lead to improvements on non-web scraped data. Our work complements these studies by exploring how ID advancements on ImageNet and related benchmarks have translated poorly to crowdsourced, global data.

## 3 MEASURING PROGRESS ON CROWDSOURCED, GLOBAL DATA

The ImageNet dataset has been an instrumental measure of progress for object recognition (Russakovsky et al., 2015). Alongside, standard ImageNet benchmarks such as ImageNet-A, ImageNet-C, and ObjectNet, have been developed to assess how well models generalize (see Section 2). However, with recent advances in foundation models such as CLIP, performance on the standard ImageNet distribution shifts benchmarks has begun to saturate (Radford et al., 2021). A key limitation of standard generalization benchmarks is that they rely on artificially induced corruptions or predefined criteria. While they represent important distribution shifts, we argue that crowdsourced, global data can provide an insightful view of model consistency in a natural setting.

### 3.1 GEOGRAPHICALLY DIVERSE DATASETS

Recently, two datasets of household objects spanning the globe were introduced: DollarStreet (Rojas et al.) and GeoDE (Ramaswamy et al., 2023). DollarStreet contains 38K images, with 96 classes, and spans 54 countries and 4 regions, while GeoDE contains 61K images with 40 classes, and spans 6 regions. Both datasets are commonly used in fairness literature to study performance disparities across images from different socioeconomic groups and regions (DeVries et al., 2019; Gustafson et al., 2023; Rojas et al.; Goyal et al., 2021; 2022; Ramaswamy et al., 2023). To study the largest catalogue of models possible, we use the ImageNet-1k class mappings released for DollarStreet and generated a similar mapping for GeoDE classes. These class mappings (see Appendix A) allow us to evaluate any vision model compatible with the original 1k ImageNet classes. *Geographically labeled datasets* such as GeoDE or DollarStreet allow us to measure generalization as it occurs in the crowdsourced data collected across geographies.

**Controlling for image quality.** Can performance differences be simply be attributed to a lack of geographic representation or regional differences in image quality? As shown in Ramaswamy et al. (2023) and Gustafson et al. (2023) both DollarStreet and GeoDE have consistent image quality and contain roughly balanced numbers of samples per region. In both datasets, images are crowdsourced and labeled by the households who took the photo. This process produces high-quality ground truth class labels.

### 3.2 MEASURING GENERALIZATION BEYOND AVERAGE ACCURACY

The most commonly reported measure of progress for standard object recognition benchmarks is the *average classification accuracy*. While using an average accuracy provides a high-level view of model performance, a holistic understanding of model reliability requires a more fine-grained evaluation. We show how using crowdsourced, global data offers a unique lens for evaluation through region subgroup disparities. In this work, we complement average benchmark accuracy with two additional metrics for assessing the rate of progress and uncovering disparities not revealed by average accuracy.

First, we are interested in measuring the rate at which each type of benchmarks (geographical or standard) benefit from advances in the field. Thus, we measure the accuracy increases for each benchmark relative to ImageNet accuracy, where the rate of progress is the slope of a linear fit. We compute the difference of progress rates between standard generalization benchmarks and geographical shift benchmarks and consider **Progress Gap** defined as:

$$\textbf{Progress Gap} := \text{Progress Rate Standard} - \text{Progress Rate Geographical} \tag{1}$$

$$= \frac{\text{Standard Improvement} - \text{Geographic Improvement}}{\text{ImageNet Improvement}}. \tag{2}$$

*Progress gap* indicates how much of the progress on standard benchmarks transfers to crowdsourced, global datasets. For example, a progress gap of 2x indicates improvements on standard benchmarks progress twice as fast as improvements on our crowdsourced benchmarks. The Progress gap measure relies on a linear fit of the accuracy trends, which are well supported by statistically significant with high Coefficients of Determination ($R^2$) seen in Table 2 (details in Appendix B).

However, there is a blind spot when using average accuracy: it may conceal poor performance on some groups relative to others (Idrissi et al., 2022). For example, a model may perform well on average, but generalize quite poorly to some regions. Fortunately, datasets with geographic labels, such as DollarStreet and GeoDE, offer an opportunity to reveal when such disparities arise.

To complement average accuracy, we propose measuring *geographic disparity* on crowdsourced datasets as more holistic measure of model reliability. For DollarStreet and GeoDE, we do so by measuring the maximum absolute difference in a model's classification performance across any two regions, which we refer to as Geographic Disparity and is defined as:

$$\Delta\textbf{Disparity} := \max\{|P_i - P_j| : i, j \in 1, \ldots, k\} \tag{3}$$

where $P_i$ indicates the performance on the $i^{th}$ region and $k$ is the number of regions. This definition can be applied broadly to any geographically labeled dataset and groupings other than regions such as country, zip code, or continent.

Progress gap, together with geographical disparity in both GeoDE and DollarStreet datasets, gives us a more holistic understanding of model reliability and progress in object recognition.

### 3.3 ASSESSING PROGRESS ON CROWDSOURCED, GLOBAL DATA

Equipped with two geographically diverse datasets and metrics of improvement, we now turn to the question: *to what extent has progress on standard ImageNet benchmarks improved generalization on crowdsourced, global datasets?* First, we compare progress rates on standard benchmarks relative to progress based on average classification accuracy of household objects around the globe (i.e. with **Progress Gap** from Equation 2). We go beyond average accuracy to probe how progress on standard benchmarks affects generalization in terms of geographic disparities (i.e. with $\Delta$**Disparity** from Equation 3) using DollarStreet and GeoDE described in Section 3.1.

We investigate a testbed of 98 models, which spans 16 architectures and includes recent foundation models such as CLIP, FLAVA, and DINOv2. We primarily use weights available in the Timm library (Wightman, 2019) for ImageNet trained models, use the OpenCLIP library for CLIP models (Ilharco et al., 2021), and use HuggingFace (Wolf et al., 2020) implementations of other foundation models. We include a comprehensive table of testbed metadata in Appendix A. Our testbed includes models trained on up to 2 billion images and with over 100 million parameters.

## 4 THERE IS A PROGRESS GAP BETWEEN STANDARD BENCHMARKS AND CROWDSOURCED, GLOBAL DATA

Here we measure the rates of progress on standard ImageNet benchmarks, along with progress on crowdsourced, global datasets. If standard benchmarks faithfully reflect object variation in natural settings, we would expect both sets of benchmarks to have consistent rates of progress. We compare the improvements on standard generalization benchmarks to crowdsourced, global benchmarks as a function of ImageNet accuracy. As shown below, we find accuracy on standard generalization benchmarks to improve by 62.75% on average, while progress on the geographically diverse DollarStreet dataset only improves by 18.9% (33.5% for GeoDE).

To isolate these progress trends, we compute linear trend lines for each benchmark. We find the trend lines are statistically significant with high Coefficients of Determination ($R^2$) as shown in

Table 2 (details in Appendix B). We discover a striking progress gap between standard generalization benchmarks and crowdsourced, global data: *progress on standard benchmarks is* $2.5\times$ *the progress on crowdsourced, global datasets*. The progress gap is consistent for both DollarStreet and GeoDE, despite these benchmarks containing different classes and collection procedures. This suggests the progress gap isn't an artifact of a particular dataset. Both the difference in progress rates, and the net improvement values point to a substantial gap in progress between the commonly reported standard benchmarks and crowdsourced, global datasets.

| Benchmark | Net Improvement (↑) | Progress Rate (↑) | **Progress Gap** | $R^2$ (↑) |
|---|---|---|---|---|
| DollarStreet (baseline) | +18.92% | 0.53 | 1.0x | 0.93 |
| *In-Distribution* | | | | |
| ImageNet-V2 | +37.74% | 1.18 | **2.2x** | 0.99 |
| *Out-Of-Distribution* | | | | |
| ImageNet-Sketch | +63.00% | 1.37 | **2.6x** | 0.75 |
| ImageNet-Rendition | +73.42% | 1.50 | **2.8x** | 0.74 |
| ObjectNet | +51.84% | 1.46 | **2.8x** | 0.90 |
| OOD Average | +62.75% | 1.44 | **2.7x** | 0.82 |

Table 2: **There is a striking progress gap between standard ImageNet benchmarks and geographic shift benchmarks**, with all benchmarks improving at *over double* the rate of DollarStreet. This translates to a net improvement on average OOD datasets that is more than *3x* the net improvement on DollarStreet. We measure progress rate as the slope of a linear fit between ImageNet accuracy and benchmark accuracy, and include the coefficient of determination ($R^2$) for each.

## 5 PROGRESS ON STANDARD BENCHMARKS EXACERBATES PERFORMANCE DISPARITIES

We found progress on crowdsourced, global data in terms of average accuracy lags considerably behind progress on standard benchmarks. While useful, average accuracy can conceal large disparities in performance indicative of poor geographic generalization. Here we address average accuracy's blind spots by studying performance disparities across regions. We measure performance disparity as the top-1 accuracy difference between the best (Europe) and least (Africa) performing regions in DollarStreet and GeoDE. We then study whether progress on standard ImageNet benchmarks improves or exacerbates geographic disparities.

### 5.1 EVEN SOTA MODELS HAVE LARGE PERFORMANCE DISPARITIES BETWEEN REGIONS

We first measure the maximum performance disparity across regions. If a model generalizes well across geographies, we would expect a small performance disparity; whereas, poor geographic generalization would lead to large disparities. We find all models have substantial disparities between regions, from ResNets to the largest CLIP models trained on 2 billion image-text pairs. In our study, ResNet models have average geographic disparities of 14.5% on DollarStreet and 5.0% on GeoDE. The best performing CLIP model actually had even more considerable disparities, with a disparity of 17.0% on DollarStreet and 6.5% on GeoDE. These considerable geographic disparities suggest average accuracy is concealing a crucial axis of generalization that remains *unsolved by today's best models*. Next, we study how progress on standard ImageNet benchmarks has affected geographic disparities.

### 5.2 PROGRESS ON IMAGENET FAILS TO RESOLVE DISPARITIES, OFTEN EXACERBATING THEM

Has progress on standard ImageNet benchmarks improved or exacerbated geographic disparities? To answer this question, we compare geographic disparities as a function of progress on ImageNet and standard generalization benchmarks. Contrary to modern intuition, we discover, as shown in Figure 2, progress on ImageNet and its generalization benchmarks not only fails to resolve geographic disparities, but actually exacerbates disparities. We find for DollarStreet *disparities between the least performant models and today's best models have more than tripled*. We also analyze performance

disparities in GeoDE finding that that improvements on standard benchmarks are not predictive of any improvement in geographic disparity (see Appendix C).

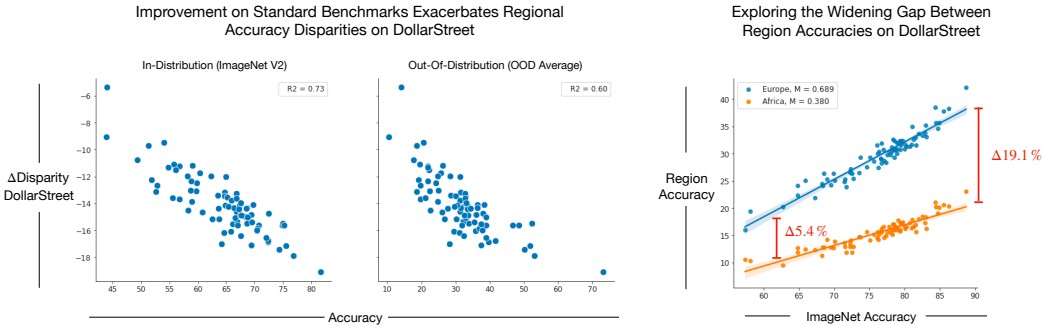

Figure 2: **Model improvement on standard ID and OOD benchmarks exacerbates the region disparity on DollarStreet**, measured as the accuracy difference between Europe and Africa subsets. Left: DollarStreet region disparity exacerbated with both ID and OOD improvements. Right: Per region accuracy on DollarStreet as a function of ImageNet accuracy shows the exacerbated region disparity.

## 5.3 EXPLORING THE WIDENING PERFORMANCE DISPARITIES BETWEEN REGIONS

To further explore the growing disparities, we isolate region performance as a function of improving ImageNet accuracy to understand individual effect on the rate of progress in each region. In Figure 2, we show accuracy in the best (Europe) and least (Africa) performing regions as ImageNet accuracy improves. While overall models also improve on each region, they improve on for Europe at almost double the rate of that for Africa, leading to a widening performance disparity between them. For GeoDE, we see much more similar rates of improvement across regions (see Appendix C).

Our analysis indicates that progress as measured by average accuracy is an incomplete picture. We find that models across architectures and datasets have large, meaningful disparities between regions, and that improvement on current benchmarks fails to improve on these disparities.

## 6 GENERALIZATION ACROSS GEOGRAPHY: OPEN CHALLENGES AND PROMISING DIRECTIONS

Next, we explore directions for improving the concerning empirical trends we uncover about model failures across geographies. We investigate multiple avenues from common robustness interventions such MixUp to scaling of both data/model size as well as forms of data curation. We find many avenues known to improve generalization on standard benchmarks fail to address generalization to crowdsourced, geographic shifts. Finally, we perform a last layer retraining (Kirichenko et al., 2022) experiment to approximate the impact of training on a more geographically representative dataset.

### 6.1 ROBUSTNESS INTERVENTIONS OFFER LIMITED IMPROVEMENTS

We evaluate popular interventions that have been shown to improve generalization on standard benchmarks: Deep AugMix, AugMix, Texture Debiasing, CutMix, and AntiAliasing ((Hendrycks et al., 2019), (Geirhos et al., 2018b), (Yun et al., 2019), (Zhang, 2019)). We evaluate these techniques using pretrained ResNet50 models. In Table 3, we show accuracy on standard benchmarks as well as geographic disparities for DollarStreet and GeoDE for models trained with each intervention compared to a baseline ResNet50 model (trained without any interventions). The majority of robustness interventions improved one benchmark's regional gap slightly, while degrading the other. The exception is AugMix, which improved the GeoDE and DollarStreet gaps by 1.86% and 0.94% respectively. Common robustness interventions overall offer limited improvements to geographic disparities, indicating a need for more holistic solutions.

| Intervention | ImageNet (↑) | OOD Avg (↑) | ∆**Disparity** GeoDE (↓) | ∆**Disparity** DS (↓) |
|---|---|---|---|---|
| Baseline | 76.34 | 30.28 | 4.96 | 15.16 |
| Deep AugMix | 76.73 | 32.92 | 5.22 | 13.53 |
| Texture Debiased | 76.73 | 31.13 | 4.70 | 16.20 |
| Ant-Aliased | 77.41 | 30.09 | 5.54 | 13.46 |
| AugMix | 77.53 | 32.51 | 3.10 | 14.22 |
| CutMix | 78.58 | 29.43 | 4.38 | 16.10 |

Table 3: Benchmarking Robustness Interventions. Most robustness interventions produced mixed results, with the exception of AugMix, which provided small improvements to geographic disparities and ImageNet accuracy. DS refers to DollarStreet.

## 6.2 FOUNDATION VISION MODELS AND SCALING

Model and data scaling have driven many recent advances (Radford et al., 2021). Here we study whether scaling's success on standard benchmarks translates to progress on crowdsourced, global datasets. We measure geographic disparity ∆**Disparity** as a function of scale in terms of data (+200 million) and model size (+100 million parameters) in Figure 3. We find *neither scaling data nor model size improves geographic disparities*. While error bars don't allow us to draw any conclusive trends, in terms of averages scaling both model and data sizes seems to exacerbate geographic disparities. We replicate the GeoDE plots in Appendix D, which contain the same relationship. We also show the scaling trend per architecture type in Appendix D, but did not find any promising scaling trends by architecture.

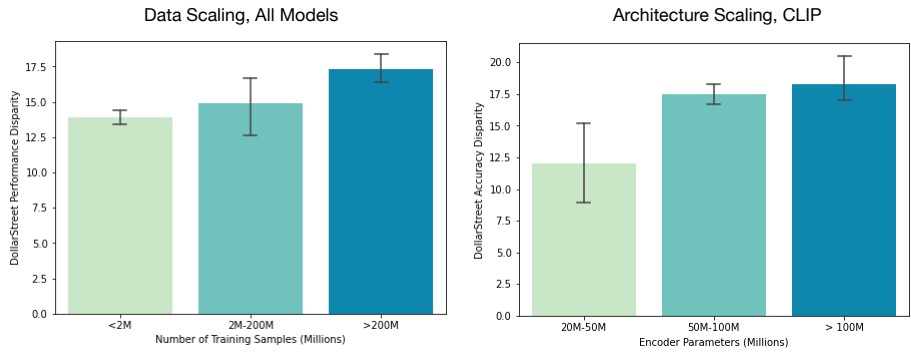

Figure 3: **Dataset and architecture scaling exacerbates region disparities on DollarStreet.**

Our results suggest scaling alone is insufficient for robustness to geographic distribution shifts. Even CLIP models have these persistent performance disparities between regions, which are not mitigated by scaling the data or architecture size.

## 6.3 THE PROMISE OF CURATING REPRESENTATIVE BALANCED DATA

Finally, we explore data curation as a promising direction for addressing geographical generalization. Prior work has highlighted data quality as a critical component of robustness improvements (Fang et al., 2022a; Idrissi et al., 2022). Recent work has also found that careful data pruning can help surpass existing performance scaling laws (Sorscher et al., 2022). In turn, we ask: to what extent can curating balanced, representative training data address geographic distribution shifts? We take a first step to answering this question by 1) analyzing the performance of DINOv2, a recent self-supervised foundation model trained with auto-curated video data, and 2) last layer retraining (Kirichenko et al., 2022) of ImageNet-pretrained ViT model on DollarStreet data. Both experiments approximate the benefits of having curated representative data for pretraining or fine-tuning stages, however, we would like to highlight that balancing web-scale data is a challenging open problem.

**DINOv2.** Despite being a mid-size model at 86 million parameters, DINOv2 achieved the smallest GeoDE region performance disparity of our testbed, with just a 2.46% accuracy difference between Europe and Africa subsets. While the model still had a significant region disparity on DollarStreet, the GeoDE improvement is remarkable for its size, and highlights that data curation offers a promising path to mitigating the tradeoff between geographic performance disparity and standard benchmarks.

**Last layer retraining on geographically representative data.** Do we need to retrain a model from scratch to reap the benefits of curated data? Inspired by Kirichenko et al. (2022) and Ramaswamy et al. (2023),

we fine-tune the last linear head of a ViT model (Dosovitskiy et al., 2020) on the training split of DollarStreet. We train the last layer for 5 epochs using Adam optimizer, learning rate $10^{-5}$ and batch size 32. We then evaluate this model on both DollarStreet and GeoDE. For GeoDE, we evaluate generalization and disparity on the subset of classes overlapping with DollarStreet classes (full details in Appendix E). We find, as shown in Table 4, last layer retraining improves average accuracy and geographic disparities on both DollarStreet and GeoDE. The average accuracy on DollarStreet's evaluation set improves by a dramatic 53.4% with geographic disparity also improving by 11.7%. Remarkably, despite retraining only on DollarStreet, we observe improvements on GeoDE of 11.5% on average accuracy and 3.2% in geographic disparities. We expand on Ramaswamy et al. (2023) by showing how last layer retraining is not just improving total performance on geographically diverse data, but improves geographic disparities, counter to our results on scaling and common robustness interventions. Our results indicate careful use of more representative data holds great promise to consistently improve both average performance and geographic disparity. We include extensions of this experiment with varying amounts of DollarStreet data in the Appendix E, as well as a rough approximation of western bias in a filtered version of LAION and a finetuning experiment in Appendix H.

**Motivating future work to curate geographically representative, web-scale training datasets.** Our analyses provide promising evidence that the concerning empirical trends we uncover could be mitigated through data. While these analyses (and mitigations presented in previous work) have shown promise, we do not present them as sufficient solutions themselves, as they are limited to the very small number of classes available in geographically diverse datasets. Our analysis suggests that these problems are not specific to architecture, training procedure, or even dataset scale. Rather, we highlight geographic bias as a common problem across web-scraped datasets, and present these analyses as motivating evidence for the field to contribute to the open research challenge of curating geographically representative, effective, web-scale training datasets.

| | Average Accuracy (↑) | | Δ Disparity (↓) | |
|---|---|---|---|---|
| | DollarStreet | GeoDE | DollarStreet | GeoDE |
| ViT | 23.46 | 65.44 | 17.12 | 4.86 |
| LLR-ViT | 76.84 $\pm$ 0.1 **(+53.41)** | 76.97 $\pm$ 0.9 **(+11.53)** | 5.47 $\pm$ 1.2 **(-11.65)** | 1.64 $\pm$ 0.6 **(-3.22)** |

Table 4: **Last layer retraining on DollarStreet improves geographic disparity and overall performance on both DollarStreet and GeoDE.** As explained in text, we report GeoDE overlapping with ImageNet. LLR-ViT refers to Last-Layer Retrained ViT.

## 7 DISCUSSION

In this work, we uncover a set of concerning empirical trends: first, that progress on crowdsourced, global data has significantly lagged behind standard benchmarks, with advances on ImageNet occurring at 2.5x the rate of progress on crowdsourced, global data. Second, we find that progress on standard benchmarks has failed to improve or exacerbated geographic disparities, with geographic disparities tripling between earliest models and today's best. We showcase the promise of using more curated and/or representative data for mitigating these trends, and emphasize curation of web-scale, geographically representative training datasets as a critical open problem for the research community (See Appendix I for discussion). Our analysis suggests that if we do not make geographic fairness an explicit desideratum in our development and benchmarking, the systems we build will continue to reinforce existing geographic disparities. We release our code and models to support future work.

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

APPENDIX

## A MEASURING GENERALIZATION ON CROWDSOURCED, GLOBAL DATA

**TestBed and Evaluation Procedure:** We include a list of the models in our testbed below, including the architecture group, evaluation type, training dataset, and the library or github source we used for model weights. For data augmentation, for all models we used the ImageNet normalization available in PyTorch, resize images to 256 pixels, and center crop to 224 pixels. Our evaluation code can be found on github at `https://github.com/facebookresearch/Geographic_Generalization`.

| Model | Architecture | Evaluation Type | Dataset | Source |
|---|---|---|---|---|
| eva-clip | CLIP | 1K | Laion-2B | Timm |
| convnext-base | ConvNext | 1K | 1K | Timm |
| convnext-large | ConvNext | 1K | 1K | Timm |
| convnext-small | ConvNext | 1K | 1K | Timm |
| dla102 | DLA | 1K | 1K | Timm |
| dla102x | DLA | 1K | 1K | Timm |
| dla169 | DLA | 1K | 1K | Timm |
| dla34 | DLA | 1K | 1K | Timm |
| dla46c | DLA | 1K | 1K | Timm |
| dla46xc | DLA | 1K | 1K | Timm |
| dla60 | DLA | 1K | 1K | Timm |
| dla60x | DLA | 1K | 1K | Timm |
| edgenet-base | EdgeNext | 1K | 1K | Timm |
| edgenet-s | EdgeNext | 1K | 1K | Timm |
| edgenet-xs | EdgeNext | 1K | 1K | Timm |
| edgenet-xxs | EdgeNext | 1K | 1K | Timm |
| hrnet18 | HRNet | 1K | 1K | Timm |
| hrnet18small | HRNet | 1K | 1K | Timm |
| hrnet30 | HRNet | 1K | 1K | Timm |
| hrnet32 | HRNet | 1K | 1K | Timm |
| hrnet40 | HRNet | 1K | 1K | Timm |
| hrnet44 | HRNet | 1K | 1K | Timm |
| hrnet48 | HRNet | 1K | 1K | Timm |
| hrnet64 | HRNet | 1K | 1K | Timm |
| lcnet100 | LCNet | 1K | 1K | Timm |
| lcnet50 | LCNet | 1K | 1K | Timm |
| lcnet75 | LCNet | 1K | 1K | Timm |
| mlpmixer | MLP | 1K | 1K | Timm |
| mlpmixerlarge | MLP | 1K | 1K | Timm |
| mobilenet-lamb100 | MobileNet-V3 | 1K | 1K | Timm |
| mobilenet-lamb50 | MobileNet-V3 | 1K | 1K | Timm |
| mobilenet-lamb75 | MobileNet-V3 | 1K | 1K | Timm |
| regnet | RegNet | 1K | 1K | Timm |
| regnet120 | RegNet | 1K | 1K | Timm |
| regnet16 | RegNet | 1K | 1K | Timm |
| regnet2 | RegNet | 1K | 1K | Timm |
| regnet32 | RegNet | 1K | 1K | Timm |
| regnet320 | RegNet | 1K | 1K | Timm |
| regnet6 | RegNet | 1K | 1K | Timm |
| regnet64 | RegNet | 1K | 1K | Timm |
| regnet8 | RegNet | 1K | 1K | Timm |
| seer1280 | RegNet | 1K | Instagram | Github[*] |
| seer320 | RegNet | 1K | Instagram | Github[*] |
| seer640 | RegNet | 1K | Instagram | Github[*] |
| regnet120x | RegNetX | 1K | 1K | Timm |
| regnet16x | RegNetX | 1K | 1K | Timm |

| | | | | |
|---|---|---|---|---|
| regnet2x | RegNetX | 1K | 1K | Timm |
| regnet320x | RegNetX | 1K | 1K | Timm |
| regnet32x | RegNetX | 1K | 1K | Timm |
| regnet4x | RegNetX | 1K | 1K | Timm |
| regnet64x | RegNetX | 1K | 1K | Timm |
| regnet6x | RegNetX | 1K | 1K | Timm |
| regnet8x | RegNetX | 1K | 1K | Timm |
| resnet101 | ResNet | 1K | 1K | Timm |
| resnet152 | ResNet | 1K | 1K | Timm |
| resnet18 | ResNet | 1K | 1K | Timm |
| resnet34 | ResNet | 1K | 1K | Timm |
| resnet50 | ResNet | 1K | 1K | Timm |
| resnet50anti | ResNet | 1K | 1K | Timm |
| resnet50augmix | ResNet | 1K | 1K | Timm |
| resnet50cutmix | ResNet | 1K | 1K | Timm |
| resnet50cutmixbaseline | ResNet | 1K | 1K | Timm |
| resnet50deepaug | ResNet | 1K | 1K | Timm |
| resnet50deepaugmix | ResNet | 1K | 1K | Timm |
| resnet50texture | ResNet | 1K | 1K | Timm |
| rexnet100 | RexNet | 1K | 1K | Timm |
| rexnet130 | RexNet | 1K | 1K | Timm |
| rexnet150 | RexNet | 1K | 1K | Timm |
| rexnet200 | RexNet | 1K | 1K | Timm |
| tinynet-a | TinyNet | 1K | 1K | Timm |
| tinynet-b | TinyNet | 1K | 1K | Timm |
| tinynet-c | TinyNet | 1K | 1K | Timm |
| tinynet-e | TinyNet | 1K | 1K | Timm |
| vgg-11 | VGG | 1K | 1K | Timm |
| vgg-13 | VGG | 1K | 1K | Timm |
| vgg-16 | VGG | 1K | 1K | Timm |
| vgg-19 | VGG | 1K | 1K | Timm |
| DINOv2 | ViT | 1K | LVD-142M | GitHub** |
| vit | ViT | 1K | 21K | Timm |
| vitlarge | ViT | 1K | 21K | Timm |
| clip-convnext-laion2b | CLIP | Zeroshot | Laion-2B | OpenCLIP |
| clip-convnext-laion2b-a | CLIP | Zeroshot | Laion-2B | OpenCLIP |
| clip-convnext-laion2b-aug | CLIP | Zeroshot | Laion-2B | OpenCLIP |
| clip-convnextlarge-laion2b | CLIP | Zeroshot | Laion-2B | OpenCLIP |
| clip-r101-openai | CLIP | Zeroshot | OpenAI | OpenCLIP |
| clip-r101-yfcc | CLIP | Zeroshot | YFCC | OpenCLIP |
| clip-r50-cc12m | CLIP | Zeroshot | CC12M | OpenCLIP |
| clip-r50-openai | CLIP | Zeroshot | OpenAI | OpenCLIP |
| clip-r50-yfcc | CLIP | Zeroshot | YFCC | OpenCLIP |
| clip-vit14-laion2b | CLIP | Zeroshot | Laion-2B | OpenCLIP |
| clip-vit14-laion400m | CLIP | Zeroshot | Laion-400M | OpenCLIP |
| clip-vit14-openai | CLIP | Zeroshot | OpenAI | OpenCLIP |
| clip-vit16-laion2b | CLIP | Zeroshot | Laion-2B | OpenCLIP |
| clip-vit16-laion400m | CLIP | Zeroshot | Laion-400M | OpenCLIP |
| clip-vit16-openai | CLIP | Zeroshot | OpenAI | OpenCLIP |
| clip-vit32-laion400m | CLIP | Zeroshot | Laion-400M | OpenCLIP |
| clip-vit32-openai | CLIP | Zeroshot | OpenAI | OpenCLIP |
| flava | PMD | Zeroshot | HuggingFace | OpenCLIP |

* The SEER Github can be found here: https://github.com/facebookresearch/vissl/tree/main/projects/SEER.
**The DINOv2 Github can be found here: https://github.com/facebookresearch/dinov2.

**Class Maps**   For DollarStreet and GeoDE datasets, we use a class mapping to ImageNet-1K to evalute 1K models, and use the original labels for DollarStreet and GeoDE to evalaute zero-shot models. We use the released mapping for DollarStreet and generate mapping for GeoDE. We generate the GeoDE mapping using the spacey model (Honnibal et al., 2020) to calculate the most similar ImageNet classes for each GeoDE class, manually selecting the most reasonable results and correcting as needed. We successfully create mappings for 36 of the 40 GeoDE classes. Below are the class mappings:

| DollarStreet Class | ImageNet Class(es) |
|---|---|
| home | manufactured home |
| street view | street sign |
| tv | television |
| washing clothes/cleaning | washing machine |
| toilet | toilet seat |
| kitchen sink | washbasin |
| drinking water | water bottle |
| stove/hob | stove |
| salt | salt shaker |
| bed | day bed |
| toys | toyshop |
| everyday shoes | running shoe |
| plate of food | plate |
| cooking pots | skillet |
| social drink | soda bottle |
| phone | cellphone |
| place where eating dinner | dining table |
| lock on front door | padlock |
| wardrobe | wardrobe |
| soap for hands and body | soap dispenser |
| ceiling | tile roof |
| refrigerator | refrigerator |
| bathroom/toilet | toilet seat |
| dish washing brush/cloth | dishrag |
| toilet paper | toilet paper |
| plates | plate |
| dish washing soap | soap dispenser |
| trash/waste | trash can |
| dish racks | plate rack |
| shower | shower curtain |
| cups/mugs/glasses | cup |
| armchair | rocking chair |
| light sources | table lamp |
| light source in livingroom | table lamp |
| books | bookcase |
| switch on/off | switch |
| light source in kitchen | table lamp |
| couch | studio couch |
| sofa | studio couch |
| roof | tile roof |
| cutlery | wooden spoon |
| cooking utensils | spatula |
| medication | medicine cabinet |
| source of cool | electric fan |
| pen/pencils | ballpoint |
| street detail | street sign |
| turning lights on and off | switch |
| music equipment | speaker |
| tools | tool kit |
| cleaning equipment | dishrag |

| | |
|---|---|
| bed kids | day bed |
| table with food | dining table |
| get water | water jug |
| paper | paper towel |
| radio | radio |
| shoes | running shoe |
| starting stove | igniter |
| freezer | icebox |
| source of heat | space heater |
| computer | desktop computer |
| jewelry | necklace |
| knifes | paper knife |
| wall clock | wall clock |
| pouring water | water jug |
| doing dishes | dishwasher |
| guest bed | day bed |
| mosquito protection | mosquito net |
| bike | all-terrain bike |
| pouring drinking water | water bottle |
| oven | stove |
| place where serving guests | eating place |
| glasses or lenses | dark glasses |
| necklaces | necklace |
| source of light | table lamp |
| parking lot | parking meter |
| waste dumps | trash can |
| eating | restaurant |
| car | passenger car |
| reading light | table lamp |
| lightsources by bed | table lamp |
| family eating | eating place |
| arm watch | digital watch |
| taking a teaspoon of salt | salt shaker |
| using toilet | toilet seat |
| sitting and watching tv | television |
| opening and closing the freezer | icebox |
| diapers (or baby-pants) | diaper |
| moped/motorcycle | moped |
| cleaning after toilet | toilet paper |
| dishwasher | dishwasher |
| opening and closing the refrigerator | refrigerator |
| answering the phone | mobile phone |
| alarm clock | analog clock |
| wheel barrow | wheelbarrow |
| listening to the radio | radio |
| dinner guests | eating place |

| GeoDE Class | ImageNet Class(es) |
| --- | --- |
| bag | backpack, purse, punching bag, sleeping bag, plastic bag, messenger bag, shopping basket, pencil case |
| hand soap | soap dispenser, lotion |
| dustbin | bucket, trash can, plastic bag, barrel |
| toothbrush | - |
| toothpaste toothpowder | - |
| hairbrush comb | - |
| chair | barber chair, folding chair, rocking chair, couch, throne |
| hat | cowboy hat, swimming cap, football helmet, poke bonnet, sombrero military hat (bearskin or shako), shower cap |
| light fixture | table lamp, spotlight, lampshade, candle |
| light switch | electrical switch |
| plate of food | plate, tray |
| spices | - |
| stove | Dutch oven, stove |
| cooking pot | frying pan, hot pot, Crock Pot, cauldron, Dutch oven, wok |
| cleaning equipment | vacuum cleaner, washing machine, mop, broom, bucket, soap dispenser |
| lighter | lighter |
| medicine | pill bottle, medicine cabinet |
| candle | candle |
| toy | teddy bear, toy store |
| jug | water jug, whiskey jug, water bottle, drink pitcher |
| streetlight lantern | torch, pole |
| front door | sliding door |
| tree | - |
| house | cliff dwelling, mobile home, barn, home theater, boathouse |
| backyard | patio |
| truck | garbage truck, semi-trailer truck, tow truck, pickup truck |
| waste container | plastic bag, trash can, barrel, bucket |
| car | garbage truck, recreational vehicle, semi-trailer truck, tow truck, sports car, railroad car, minivan, station wagon, minibus, jeep, limousine, taxicab, convertible, pickup truck moving van, police van, race car |
| fence | chain-link fence, picket fence, split-rail fence |
| road sign | traffic or street sign |
| dog | Bernese Mountain Dog, Sealyham Terrier, Toy Poodle, toy terrier, African wild dog, husky, Maltese, Beagle, Labrador Retriever, Cairn Terrier, dingo, Australian Kelpie German Shepherd Dog, Golden Retriever, Malinois, Norwegian Elkhound, Chihuahua, Tibetan Mastiff, Staffordshire Bull Terrier, American Staffordshire Terrier Pembroke Welsh Corgi, Miniature Poodle, Basenji, Rhodesian Ridgeback, Appenzeller Sennenhund, Ibizan Hound |
| wheelbarrow | wheelbarrow |
| religious building | mosque, church, monastery, bell tower, altar |
| stall | - |
| boat | motorboat, canoe, fireboat, lifeboat, sailboat, submarine, ocean liner, trimaran, catamaran |
| monument | triumphal arch, obelisk, stupa, pedestal, brass memorial plaque, megalith |
| flag | flagpole |
| bus | minibus, school bus, trolleybus |
| storefront | grocery store, tobacco shop, bookstore, toy store, barbershop, candy store, shoe store |
| bicycle | tricycle, mountain bike, tandem bicycle, unicycle |

**Benchmark Use**    Our analysis relies on two benchmarks, DollarStreet and GeoDE, to characterize geographical disparity, and our results (as with all research analyses), should be interpreted considering the datasets used.  However, we emphasize that these benchmarks are reliable measures of geographical disparities, as they are 2-6x larger than existing robustness benchmarks, and are supported by a wealth of research use in fairness (DeVries et al., 2019; Gustafson et al., 2023; Rojas et al.; Goyal et al., 2021; 2022; Ramaswamy et al., 2023). DollarStreet and GeoDE have 61K and 38K respectively, compared to 10K in ImageNet-A and 30K samples in ImageNet Sketch.

There are some significant differences between these benchmarks, which we discuss below: DollarStreet is more diverse in terms of geography and household income. DollarStreet was curated to explicitly capture a broad set of households, sending photographers to over 63 countries to people's homes, and selecting households with a variety of incomes. GeoDE was crowdsourced among internet users from 19 countries and was designed to show the importance of crowdsourced data compared to internet scraped data. GeoDE has a 1-to-many label mapping, whereas DollarStreet has a 1:1 label mapping (but has several original labels). As shown below, GeoDE classes are coarser as there are GeoDE classes that map to as many as 25 ImageNet classes, while DollarStreet has 1 ImageNet label for each DollarStreet label.  This makes GeoDE an easier dataset than DollarStreet for ImageNet models (we also evaluate a large number of zero-shot models that use the ground truth labels, without requiring mapping).

Despite the large differences in dataset curation and difficulty, we find very similar results on DollarStreet and GeoDE, indicating that the problems we discover are not specific to a given dataset curation, or labeling. We find that both datasets have large geographic disparities, even with SOTA models. Most critically, we find for both benchmarks that disparities are not resolved by progress on standard generalization benchmarks, dataset scaling, standard robustness interventions, or architecture scaling.

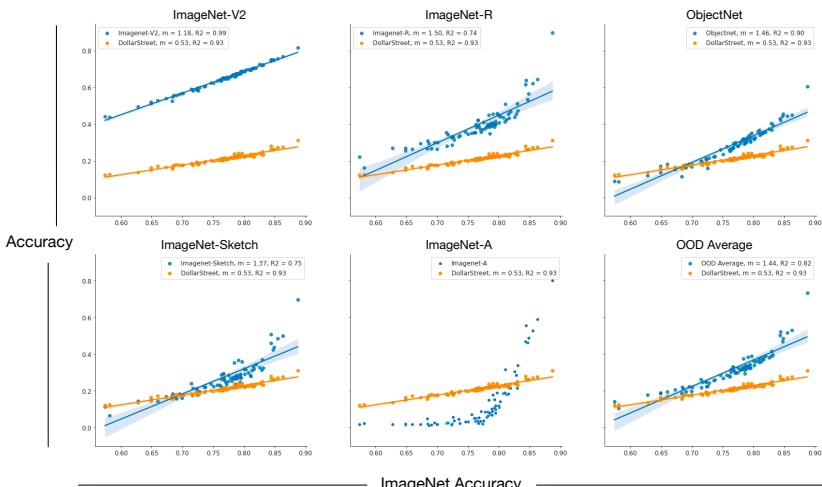

Figure 4: Progress on each benchmark (blue) as a function of ImageNet, compared to DollarStreet (orange).

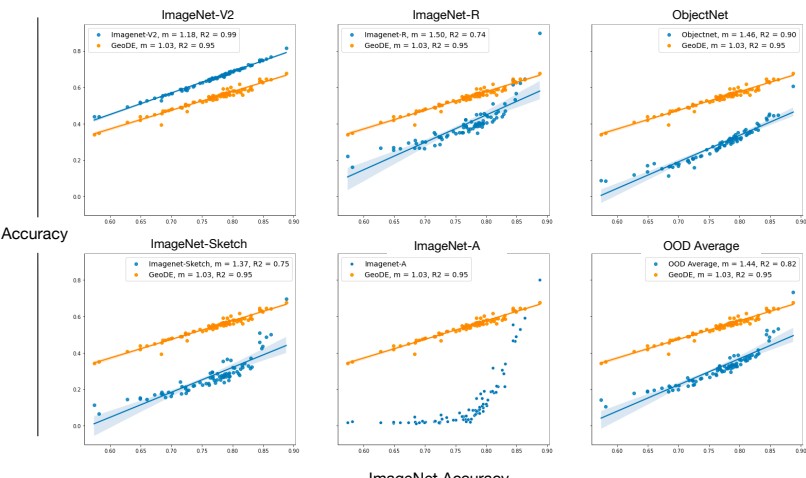

Figure 5: Progress on each benchmark (blue) as a function of ImageNet accuracy, compared to GeoDE (orange).

## B THE PROGRESS GAP BETWEEN STANDARD AND CROWDSOURCED, GLOBAL GENERALIZATION DATASETS

In Figure 4 and Figure 5 we show the performance on each standard ImageNet benchmark as a function on ImageNet performance, comparing the progress rates with DollarStreet and GeoDE respectively.

## C PERFORMANCE DISPARITIES

We show the GeoDE version of Figure 2 below in Figure 7, finding that improvement on standard imagenet benchmarks does not significantly impact regional accuracy disparities on GeoDE. We also show the relationships between Europe and Africa subsets of DollarStreet and GeoDE individually in Figure 6 and Figure 8.

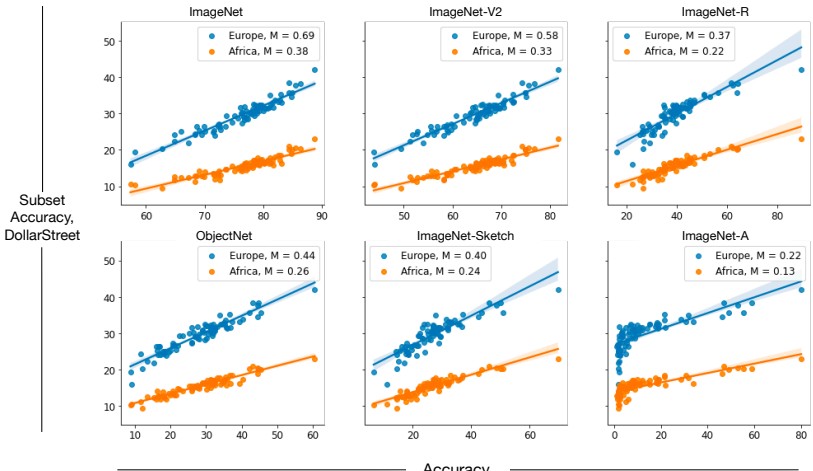

Figure 6: **Model improvement on both in-distribution and out-of-distribution benchmarks exacerbates the region disparity on DollarStreet**. Region disparity is measured as the accuracy difference between Europe and Africa subsets.

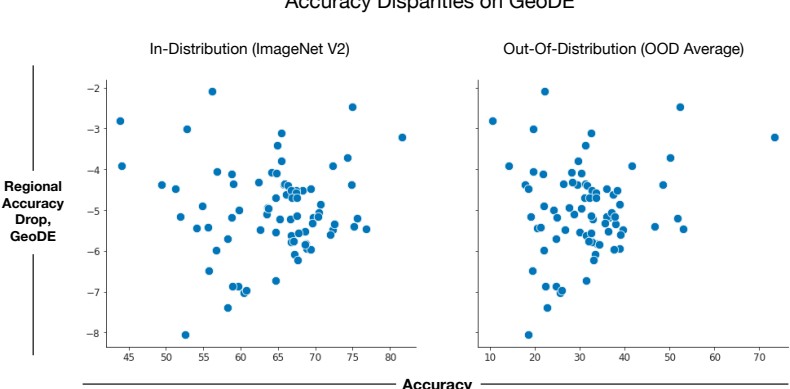

Figure 7: **Model improvement on both in-distribution and out-of-distribution benchmarks fails to improve the region disparity on GeoDE**. Region disparity is measured as the accuracy difference between Europe and Africa subsets.

| DS Income Quartile | Europe Acc (%) | Africa Accuracy (%) | Region Disparity (%) |
|---|---|---|---|
| Q3 | 51.9 | 45.2 | 6.6 |
| Q2 | 61.1 | 52.5 | 8.7 |

Table 8: **There are significant performance disparities across Region, when controlling for Income Quartile in DollarStreet. Accuracy numbers from OpenAI's ViT32 CLIP model.**

# D  FOUNDATION MODELS AND SCALING

We replicate the plots in Figure 3 for GeoDE in Figure 10.

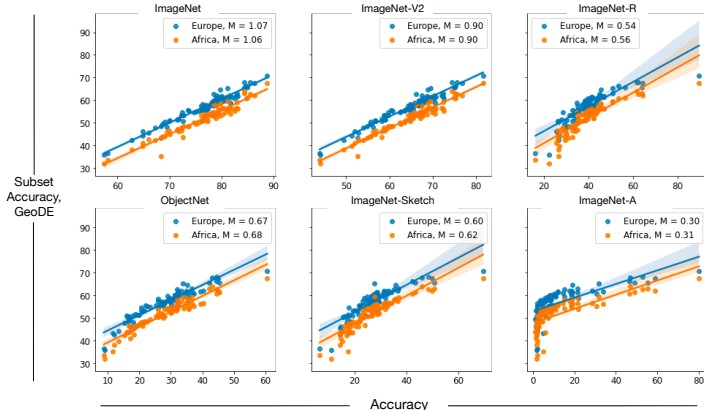

Figure 8: Model improvement on both in-distribution and out-of-distribution benchmarks does not improve the region disparities on GeoDE.

### Replication of Figure 8 for Other Region Pairs

Figure 9: **Similar trends occur with other region subset relative to the best performing subgroup (Europe), where there is a lower progress rate compared to the European subset.**

## E  REPRESENTATIVE DATA

The GeoDE classes with overlapping ImageNet labels of DollarStreet include: hand soap, dustbin, chair, light fixture, light switch, plate of food, stove, cooking pot, cleaning equipment, lighter, medicine, toy, jug, house , waste container, car, road sign, wheelbarrow, storefront, bicycle.

## F  LAION CLUSTERING EXPERIMENTS

In order to estimate the regional distribution of LAION, we perform an experiment to cluster samples by their relation to DollarStreet images in the embedding spaces. We use OpenCLIP's model (trained on LAION400M) to generate embeddings for DollarStreet images according to West (Europe, Americas) and Non West (Asia, Africa) categories. We then use KNN (testing a range from k=3 to k=113) to evaluate a filtered version of LAION. Our results approximate almost 60% of LAION samples are from western regions, and strikingly, that Europe has 2.24X more images represented

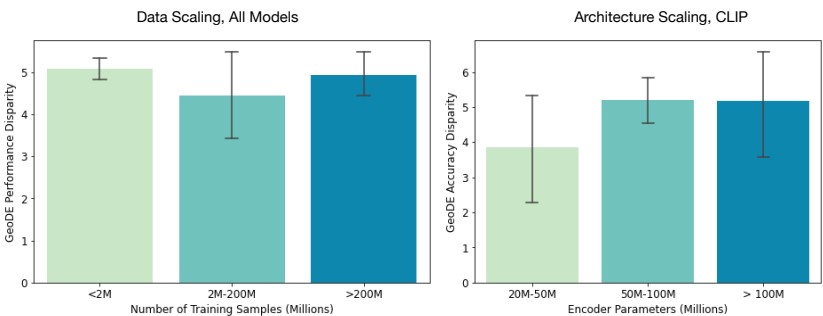

Figure 10: **Dataset and architecture scaling fails to reduce region disparities on GeoDE.**

than Africa. Our results very closely match previous work conducted on ImageNet and OpenImages **?** This analysis further sheds light on the pervasiveness of western bias in large, web-scale datasets used to train SOTA models.

## G  EXTENSION OF LAST LAYER RETRAINING EXPERIMENTS

| DS Subset (%) | DS Avg Acc (%) | DS Disp. (%) | GeoDE Avg Acc (%) | GeoDE Disp. (%) |
|---|---|---|---|---|
| 5 | $52.79_{\pm 1.52}$ | $9.20_{\pm 0.49}$ | $71.51_{\pm 2.98}$ | $3.97_{\pm 0.60}$ |
| 10 | $62.13_{\pm 0.58}$ | $5.93_{\pm 0.86}$ | $77.87_{\pm 1.32}$ | $4.00_{\pm 0.73}$ |
| 25 | $69.50_{\pm 0.30}$ | $3.27_{\pm 0.40}$ | $75.77_{\pm 1.09}$ | $3.25_{\pm 1.50}$ |
| 50 | $73.51_{\pm 0.81}$ | $3.11_{\pm 0.20}$ | $76.20_{\pm 1.89}$ | $2.59_{\pm 0.85}$ |
| 75 | $75.71_{\pm 0.12}$ | $2.69_{\pm 0.62}$ | $76.23_{\pm 0.32}$ | $3.82_{\pm 0.33}$ |
| 100 | $76.90_{\pm 0.15}$ | $1.82_{\pm 0.70}$ | $76.56_{\pm 1.15}$ | $4.29_{\pm 0.78}$ |

Table 9: **Last layer fine-tuning experiment with varying amounts of DollarStreet training data.**

## H  APPROXIMATING THE IMPACT OF WESTERN DATASET BIAS ON PERFORMANCE DISPARITY THROUGH DOLLARSTREET FINETUNING

To explore the relationship between western training data and Europe-Africa performance disparities, we conduct a finetuning experiment with varying proportions of Western DollarStreet data. We use an ImageNet pretrained ViT from Timm (vit_base_patch16_224), finetuning it on the DollarStreet training dataset that is subsampled to have increasing proportions of Western data (from 50% western, which is balanced, to 100% western, which uses only the western subset). We control for the total number of samples, and sample with a common random seed. Given the small size of the finetuning dataset, we use a smaller learning rate of 0.00001, training for 30 epochs, which did not show significant evidence of overfitting. We find in Figure 11 that increasing the Western bias in the finetuning dataset results in significantly larger performance disparities in the resulting model. We hope that this experiment provides more evidence that training dataset imbalance likely plays a significant role in the performance disparities we uncover in this work.

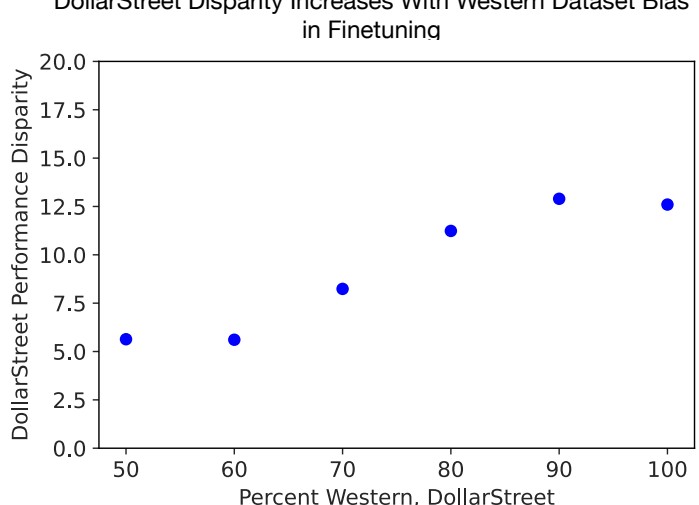

Figure 11: **Europe-Africa Performance Disparity doubles with increasingly Western dataset distributions during Finetuning.**.

## I    IMPLICATIONS FOR DATA SELECTION AND CURATION

Our work has several implications for dataset evaluation and benchmarking:

- **Geography as an important axis of generalization**. Our empirical evidence indicates that geographical diversity is a missing axis from standard benchmarks, yet geographic shifts remain a pervasive failure mode of existing models. Our work suggests the need for more diverse datasets with data collected from multiple regions. The development of these benchmarks and training datasets may require more detailed curation efforts than standard webscraping. We hope that our empirical findings provide ample motivation for such work.

- **Designing Evaluation Benchmarks**: Our work provides evidence that data sources, label curation, geographic representation, and income information should all be closely considered. In this work, we focus on geographic representation, providing some evidence that these disparities are data driven. As a result, we believe our work highlights the need for for geographically diverse benchmarks, including datasets that are grounded in real-world use. This is often a significant advantage of crowdsourced data, which can be collected to better reflect use-cases than internet-scraped datasets.

- **Designing Precise, Useful Metrics** Our work also highlights the utility of analyzing subgroups within generalization benchmarks, and the limitations of analyzing an average accuracy. By analyzing regional subgroups, we were able to discover disparate progress trends in the field which are unsolved by scaling and standard robustness intereventions. We believe that this analysis approach could similarly be used in other subgroup contexts, and hope that our work motivates future work looking more closely at how models improved (the kinds of accuracy gains made, improvement on which portions of data). Particularly, we hope that our progress gap measure will be analyzed with respect to other important subgroups in relevant datasets, which can ensure that accuracy improvements improve reliability across groups.

Overall, we suggest that there is no one particular benchmark or tool that can perfectly measure model reliability, but that having a comprehensive suite of measures, and by analyzing subgroups more precisely, we can improve our understanding of model behavior and its reliability.

## J    DRIVING MECHANISMS OF GEOGRAPHICAL PERFORMANCE DISPARITIES

Our work primarily focuses on highlighting geographical disparity as an understudied and open research challenge for the community by providing comprehensive empirical evidence that this problem is pervasive across architectures and training paradigms. While the exact underlying mechanisms for models' disparities across geographies (as well as a solution to this problem) is an open question, below we outline our main hypotheses for the driving reasons behind these disparities:

- **Dataset imbalance**: Our work shows that the geographical disparities are persistent across 100+ models with a variety of training architectures and paradigms. This suggests that the disparities are driven by geographical data imbalance in pretraining datasets. Moreover, prior works showed ImageNet is heavily western biased Shankar et al. (2017a), and in our paper Appendix F we present a kNN clustering experiment, where we analyze CLIP embeddings of LAION and DollarStreet to approximate the regional distribution of LAION. We find that our proxy measure indicates an extreme imbalance in LAION, which aligns with existing work on ImageNet. Additionally, in Section 6.3 we show that fine-tuning the last layer of a ViT model on DollarStreet improves disparities on **both** DollarStreet and GeoDE, showing the promise of interventions using geographically diverse data.

- **Distribution shifts and factors of variations between geographies**: Gustafson et al Gustafson et al. (2023) annotated factors of variation in DollarStreet and found that variations in texture, occlusion, and lighting were among the factors most associated with performance disparities. Models pre-trained on ImageNet may overrely on these factors and other spurious features such as backgrounds Xiao et al. (2020) which also contributes to geographical disparities.

- **Household income**: In addition to geographical labels, DollarStreet also has income group labels, which is correlated with geography. We hypothesize that the lower income group is underrepresented in pre-training data and also presents a distribution shift in terms of factors of variations discussed above.

Importantly, scaling up model or dataset size as well as standard robustness interventions such as data augmentation, while being effective on standard ImageNet OOD benchmarks, don't lead to improvements in geographical generalization, highlighting the unique challenges of geographical distribution shifts. We hope that our empirical insights motivate future work into these driving mechanisms.

