# OpenReview forum: "Does Progress On Object Recognition Benchmarks Improve Generalization on Crowdsourced, Global Data?"
_ICLR.cc/2024/Conference — ICLR 2024 poster_

### Official Review · Reviewer_NCcf · 2023-10-31

**Soundness:** 3 good
**Presentation:** 4 excellent
**Contribution:** 3 good
**Rating:** 6
**Confidence:** 4

**Summary:**

Evaluating ~100 ImageNet models across different architectures, scales and pretraining datasets, the authors show that progress on ImageNet and ImageNet-adjacent generalization benchmarks is faster than on geographically diverse data. The crowdsourced, global data they consider are the DollarStreet and GeoDE datasets (released in prior work). They demonstrate that the geographic disparity (difference between the accuracy on European and African objects) increases as the models get better on ImageNet. They show that different robustness interventions and scaling the models and the datasets offer limited improvements, while training and fine-tuning on more diverse and carefully curated balanced data offers a path forward.

**Strengths:**

* The paper is well-written and easy to understand.
* Studying the generalization ability and geographic/socioeconomic disparities of computer vision models is an important topic.
* This work conducts a large-scale evaluation (of ~100 models) on the 2 geographically diverse datasets exploring the trends of ImageNet models on them.
* There are 2 key findings which I find interesting:
    * Models progress faster on standardized benchmarks than on the geographic datasets. This makes sense given that the standardized benchmarks are better aligned to ImageNet -- the dataset on which the evaluated models were trained and/or fine-tuned.
    * Better ImageNet models increase the geographic disparities as measured on the evaluated datasets. This finding was intuitive and understandable, but worth emphasizing and pointing out to me.
* The authors promise that they will release their code in an easy to use manner ("with just 4 lines of code").
* I appreciate that the authors present a balanced view on the significance of geographic distribution shifts. In particular, they do not see it as an ultimately universal metric but propose it as an additional metric that should be tracked and evaluated.

**Weaknesses:**

While the findings of this work are interesting and worthwhile, I feel that the technical contributions are relatively limited. In particular, I believe that better attribution to concurrent/prior works and highlighting the consistencies with them might be helpful and fair, especially in the cases where the experimental design is inspired by them.

1. The experiment in Sec. 5.1. is consistent with prior/concurrent work: Rojas et al., DeVries et al. and Ramaswamy et al.
2. The experiment in Sec. 6.3. is also aligned with prior work. Rojas et al. find that training on DollarStreet "can improve the performance of classification tasks for items from lower-income homes", while Ramaswamy et al. "extract features using a ResNet50 model ... and retrain the final layer".

Meanwhile, I acknowledge that, in contrast to prior work, this paper performs a large scale study (with ~100 models) and examines the effectiveness of fine-tuning on one dataset (DollarStreet) and evaluating on another (GeoDE). However, the benefits of the later are also somewhat limited, as we can see in Table 8 - the results on GeoDE are similar regardless of the portion of DollarStreet training data (going from 10 to 100%).

Minor: sometimes the legends and axes on the figures, e.g., on figs. 2 and 4 and 5 in the appendix are a bit hard to read.

**Questions:**

Q1: I could not find any discussion on the differences between the DollarStreet and GeoDE datasets, while some of the results and the trends on them sometimes differ. Are there any fundamental differences between the two datasets?
* Why is GeoDE omitted from Table 2. Rate of change on it seems closer to ImageNet-V2 and ImageNet-Sketch for example, compared to rate of change on DollarStreet.
* Comparing Figs. 4 and 5 in App. B, the accuracies on GeoDE are significantly higher than those on Dollar Street and sometimes even higher than the ImageNet induced benchmarks. Why is that the case? Is GeoDE "easier" because there are fewer labels or is there another fundamental reason?
* Do you have a hypothesis why the geographic disparities are much more consistent (relatively constant with the model improvements on ImageNet) for GeoDE compared to Dollar Street (Figure 8)? Why are the datasets different in that matter?

Q2: I would like to better understand the driving source of the disparities on Dollar Street. Is it the geographic region or the family income? I expect that these two are correlated, but could you please provide any quantitative results about the disparity when also controlling for family income? I.e., instead of comparing the groups Europe and Africa, compare the groups (Europe, some income level) and (Africa, the same income level).

Q3: Is "Progress Gap" a fair or meaningful metrics? E.g. (following-up on Q1), geographic datasets may be easier/more difficult, with different labeling biases, etc. Could you please provide a brief discussion what might influence the progress rates / contribute to its differences across datasets and further motivate the usage of the progress gap metric?

Q4: Should we expect the models to have high accuracy on all regions? E.g., would fine-tuning w.r.t region deployment be a good idea in certain use cases? How do you envision the models should be deployed in practice? If we look at the problem from fairness perspective, it has been well-known that there exists a trade-off between fairness and accuracy in general, so it might make sense to have different models for the different regions.

---

> ### Author Response · Authors · 2023-11-16
>
> Thank you for your appreciation of our work, and thoughtful questions. We are glad you found our work to address an important topic, containing interesting and important findings, providing a balanced view, and well-written. We hope to have addressed your questions, and welcome any additional questions or suggestions on how we can strengthen our work:
>
> **Difference Between DollarStreet and GeoDE and Figure 8**:
>
> There are a few main drivers of the dataset differences from our perspective:
> 1. DollarStreet is more diverse in terms of geography and household income. DollarStreet was curated to explicitly capture a broad set of households, sending photographers to over 63 countries to people’s homes, and selecting households with a variety of incomes. GeoDE was crowdsourced among internet users from 19 countries and was designed to show the importance of crowdsourced data compared to internet scraped data.
> 2. GeoDE has a 1-to-many label mapping, whereas DollarStreet has a 1:1 label mapping. As shown in Appendix A.1 , GeoDE classes are coarser as there are GeoDE classes that map to as many as 25 ImageNet classes, while DollarStreet has 1 ImageNet label for each DollarStreet label. This makes GeoDE an easier dataset than DollarStreet for ImageNet models (we also evaluate a large number of zero-shot models that use the ground truth labels, without requiring mapping).
>
> Despite the large differences in dataset curation and difficulty, we find very similar results on DollarStreet and GeoDE, indicating that the problems we discover are not specific to a given dataset curation, or labeling. We find that both datasets have large geographic disparities, even with SOTA models. Most critically, we find for both benchmarks that disparities are not resolved by progress on standard generalization benchmarks, dataset scaling, standard robustness interventions, or architecture scaling.
>
> **GeoDE in Table 2**:
>
> We included only DollarStreet in Table 2 in order to avoid confusion in reader interpretation of our Progress gap measure, which is a ratio of the rate between the benchmarks with respect to one baseline (DollarStreet). We find similar trends hold, showing conclusions are consistent across both datasets - the progress rate on GeoDE is smaller compared to other benchmarks. We have these numbers in Figure 5 in the Appendix and would be happy to add them to the main table if helpful.
>
>
> **Income Variation in DollarStreet**:
>
> We agree that controlling for income variation is very interesting, thank you for suggesting this analysis! We have run an analysis of OpenAI’s CLIP trained on LAION 2B, analyzing the per-region accuracies within each income quartile. For the middle two quartiles (the only quartiles with both Europe and Africa data), we found that there was still a substantial performance drop across regions. This indicates that even when controlling for income, there is a significant performance degradation across geographies. We have included the table in Appendix C, and below for reference - thank you for the excellent suggestion, and we welcome any further advice to clarify this point.
>
> | DS Income Quartile | Europe Avg Acc (\%) | Africa Average Accuracy (\%) | Region Disparity (\%) |
> |--------------------|---------------------|------------------------------|:---------------------:|
> | Q3                 | 51.9                | 45.2                         | 6.6                   |
> | Q2                 | 61.1                | 52.5                         | 8.7                   |

---

> > ### Author Response · Authors · 2023-11-16
> >
> > **Utility of Progress Gap Metric Across Datasets**:
> >
> > We completely agree that, along with any evaluation metric, our Progress Gap metric is best understood when considering the datasets used. However, we have defined our Progress Gap metric to be a generalizable and reliable measure. The progress gap will be a useful measure as long as 1) the accuracy trend is accurately described as linear, and 2) the labels are accurate. For current standard benchmarks, these conditions hold easily. We see very high R2 values for our linear fits (see Table 2), and the linear fit of this trend has been also shown in accuracy on the line works [1].  as can be shown in our high R2 values of our linear fit in, but we include some language in Section 3.2 clarifying these conditions when we introduce the metric. Thank you for the excellent suggestion to broaden the impact of our work, and we hope this addresses your question.
> > For DollarStreet and GeoDE specifically, we emphasize that the benchmarks are reliable measures of geographical disparities, being 2-6x larger than existing robustness benchmarks and having labels that are grounded in real-world use and supported by a wealth of research use.
> >
> > 1. John P Miller, Rohan Taori, Aditi Raghunathan, Shiori Sagawa, Pang Wei Koh, Vaishaal Shankar, Percy Liang, Yair Carmon, and Ludwig Schmidt. Accuracy on the line: on the strong correlation between out-of-distribution and in-distribution generalization. International Conference on Machine Learning, pages 7721–7735. PMLR, 2021.

---

> > > ### Comment · Reviewer_NCcf · 2023-11-21
> > > **Response to the authors**
> > >
> > > I thank the authors for their extensive rebuttal! I do believe the paper is now more complete.
> > >
> > > However, after looking at the GeoDE paper again (Ramaswamy et al.), I notice that they also perform an experiment where they train on GeoDE and confirm that this also improves the results on DollarStreet (demonstrating the transferability between the two datasets). Thus, I believe the findings in Section 6 are unsurprising or broadly confirm prior work. After reading the other reviews and responses to them as well, I would like to keep my score.
> > >
> > > (I have a final minor suggestion about the formatting of the appendix: placing the tables and the figures closer to the text that refers to them would make the navigation easier)

---

> > > > ### Author Response · Authors · 2023-11-21
> > > >
> > > > Thank you for your response! We are glad you appreciate our efforts, and believe they have improved our work! We apologize for not making our work’s distinction clear in the draft, we hope to further clarify below and with added language in the paper (highlighted in related works, and in Sec 6.3). Please let us know if these distinctions make our contributions more clear:
> > > >
> > > > **Experiments in Ramaswamy et al:**
> > > >
> > > > Ramaswamy et al, perform last layer retraining on an ImageNet pretrained ResNet50 model with a mix of GeoDE and ImageNet, and show that doing so improves model per-region accuracy on GeoDE (Sec 6.1, Table 6). They show how this model improves class accuracy for the 10 out of the 96 classes in DollarStreet that overlap with GeoDE (Sec. 6.1, Table 7).
> > > >
> > > > **Experiment 6.3 in Our Work**:
> > > >
> > > > Our work studies *regional disparities*, and our contribution is to study these disparities in the context of the field’s progress, including as a function of scaling, benchmark progress, and recent interventions. Our experiment in Sec 6.3 shows the impact of last-layer retraining *on the geographical performance disparities* found in DollarStreet and GeoDE, which to our knowledge has not been reported directly in prior work. While Ramaswamy et al show how last layer retraining on GeoDe can improve a model’s per-class accuracy on 10 DollarStreet classes, **their experiment fails to answer our central question, which is asking if last layer retraining can improve the region disparity**.
> > > >
> > > > Our general goal is to highlight how **improving regional disparity is not simply a matter of improving total accuracy**. We see this in Fig. 2 of our paper, where despite progress on the DollarStreet dataset in total, the regional disparity has not only not been improved, but has been *exacerbated* over time. We hope that our last layer retraining experiment provides evidence that, in contrast to our results for scaling and ImageNet improvements, last layer retraining can practically improve regional disparities across datasets. We also hope it motivates future work in larger-scale data curation by providing evidence that training on a more geographically representative dataset can not only improve total performance, but improve the geographical fairness of the model.
> > > >
> > > > We believe that these regional disparities have important consequences for the field. Our work uncovers an alarming trend, finding geographic disparity as a pervasive problem unsolved by standard benchmark progress, scaling, or common interventions, and worsening over time. We hope that our work demonstrates the impact of this problem, and provides both motivation and insight for new directions of research in this area, including in careful data rebalancing and in design of specific geographic mitigations.
> > > >
> > > > (Thank you also for your appendix formatting suggestion, we have changed the formatting in our revision to make navigation easier).

---

### Official Review · Reviewer_PnXg · 2023-11-01

**Soundness:** 3 good
**Presentation:** 3 good
**Contribution:** 3 good
**Rating:** 8
**Confidence:** 4

**Summary:**

The paper conducts an extensive empirical evaluation of many vision models with a focus on understanding their geographical robustness. They choose two crowdsourced, geographically distributed datasets GeoDE and DollarStreet to highlight that progress on standard robustness benchmarks curated from ImageNet need not necessarily indicate geographically equitable progress. Specifically, they highlight that improvements in backbones, architectures and data showcase gains in standard benchmarks but progress on crowdsourced datasets is much more sluggish. Secondly, they also highlight the discrepancy between gains in western-centric data (European) compared to Afro-Asian data. Finally, they allude to few possible future directions which could help alleviate this bias.

**Strengths:**

- This paper addresses a very pertinent problem of investigating the equity of progress through modern deep learning advances in computer vision. They make several important observations pertaining to poor geographical robustness of current model, which can spur several future directions.

- The authors cover several key advances in their study, including large data, larger architecture and different architectures and the observations hold for most cases.

- A brief analysis of possible future directions is also presented, although it is notable that real progress demands labeled data from under-represented geographies, so unlabeled generalization is still an open challenge.

**Weaknesses:**

- The paper really does not answer the questions as to why there is a disparity between progress in disparate geographical groups. Is it due to the dataset bias in training data? Is it because of the domain shifts/ label shift between different geographies? Although the current observation that there is a accuracy gap is important, it would be more useful if an inisght into the possible causes is also presented.

- The authors note that they perform manual re-labeling of DollarStreet and GeoDA categories to ImageNet classes. Does this induce any kind of labeling noise which might explain the result? For example, images percieved by model to be `stove` are actually evaluated with `oven`. Since oven and stove designs change worldwide, could this be a possible reason for the accuracy drops? Adding to this, can you also choose only these classes from other robustness benchmarks as well? (Like select a subset of ImageNet-S with only those labels which were mapped to DollarStreet and then evaluate the accuracy). This somehow seems a more fair comparison.

- Adding to the above point, it would be useful to include another contemporary dataset GeoNet [1] into the evaluation since they seem to directly source their dataset based on labels from the ImageNet (thus avoiding relabeling).

- Several recent efforts to study the problem of geographical adaptation are not cited or compared [1,2,3]. GeoNet seems to be a relevant work which addresses similar issues with broadly similar observations. Specifically, the conclusion that scaling model sizes and datasets would not automatically yield geographical robustness seems related, but no discussion or comparison with this work is presented.

- The disparity difference is only computed against Europe and Africa. Does the observations also hold for, say, Europe and Asia?

- Sec 5.3: Could the authors hypothesize why rates of improvement across regions in GeoDE is much more uniform compared to DollarStreet?

[1] Kalluri, Tarun, Wangdong Xu, and Manmohan Chandraker. "GeoNet: Benchmarking Unsupervised Adaptation across Geographies." CVPR. 2023.

[2] Prabhu, Viraj, et al. "Can domain adaptation make object recognition work for everyone?." CVPR. 2022.

[3] Yin, Da, et al. "GIVL: Improving Geographical Inclusivity of Vision-Language Models with Pre-Training Methods." CVPR. 2023.

**Questions:**

The major question to the authors is their opinion on why the noted differences are observed. I am eagerly waiting for the responses on this and several other clarifications requested above, and I would be happy to further raise my rating.

---

> ### Author Response · Authors · 2023-11-16
>
> Thank you for your thorough engagement with our work and thoughtful suggestions! We are particularly glad that we were able to convey the very pertinent problem of geographical disparities, and are glad that you found the insights in our work important. We hope we addressed your questions thoroughly below, and welcome any additional questions!
>
> **Driving Mechanism of Geographical Disparity**:
>
> Our work primarily focuses on highlighting geographical disparity as an understudied and open research challenge for the community by providing comprehensive empirical evidence that this problem is pervasive across architectures and training paradigms. While the exact underlying mechanisms for models’ disparities across geographies (as well as a solution to this problem) is an open question, below we outline our main hypotheses for the driving reasons behind these disparities:
>
> 1. **Dataset imbalance:** Our work shows that the geographical disparities are persistent across 100+ models with a variety of training architectures and paradigms. This suggests that the disparities are driven by geographical data imbalance in pretraining datasets. Moreover, prior works showed ImageNet is heavily western biased [1], and in our paper Appendix F we present a kNN clustering experiment, where we analyze CLIP embeddings of LAION and DollarStreet to approximate the regional distribution of LAION. We find that our proxy measure indicates an extreme imbalance in LAION, which aligns with existing work on ImageNet. Additionally, in Section 6.3 we show that fine-tuning the last layer of a ViT model on DollarStreet improves disparities on **both** DollarStreet and GeoDE, showing the promise of interventions using geographically diverse data.
>
> 2. **Distribution shifts and factors of variations between geographies:** Gustafson et al [2] annotated factors of variation in DollarStreet and found that variations in texture, occlusion, and lighting were among the factors most associated with performance disparities. Models pre-trained on ImageNet may overrely on these factors and other spurious features such as backgrounds [3] which also contributes to geographical disparities.
> 3. **Household income:** In addition to geographical labels, DollarStreet also has income group labels, which is correlated with geography.  We hypothesize that the lower income group is underrepresented in pre-training data and also presents a distribution shift in terms of factors of variations discussed above.
>
> Importantly, scaling up model or dataset size as well as standard robustness interventions such as data augmentation, while being effective on standard ImageNet OOD benchmarks, don't lead to improvements in geographical generalization, highlighting the unique challenges of geographical distribution shifts. We add this discussion as a section in Appendix I.
>
> [1] Shankar et al, No Classification without Representation: Assessing Geodiversity Issues in Open Data Sets for the Developing World
> [2] Gustafson et al, Pinpointing Why Object Recognition Performance Degrades Across Income Levels and Geographies
> [3] Xiao et al, Noise or Signal: The Role of Image Backgrounds in Object Recognition
>
> **Label Mappings and Label Noise**:
>
> Thank you for the insightful question! We also considered label noise as a potential concern, which is why we explicitly conduct zero shot classification experiments, where we prompt a model with the original DollarStreet labels rather than relying on an ImageNet mapping. Specifically, we evaluate a suite of CLIP models (See Appendix A for list). We find zero shot models have consistent geographic disparities on both DollarStreet and GeoDE benchmarks, finding some of the largest disparities for the largest CLIP models, even with scaling (see Figure 2). We would also like to emphasize that in both cases with DollarStreet and GeoDE, the original labels given are much more coarse than ImageNet (for example: ‘bag’ label compared to ‘purse, pencil case, backpack, etc.’), thus when performing label mapping to ImageNet-1K each example DollarStreet and GeoDE is assigned **multiple** plausible ImageNet labels. This adds additional protection against label noise, as the model's prediction is considered correct if it falls within any of the more granular ImageNet categories. We also added a replication of Figure 5 with DollarStreet’s Top 5 accuracy (rather than Top 1), finding the same result: that our rate of improvement on DollarStreet lags significantly behind progress on standard generalization benchmarks.

---

> > ### Author Response · Authors · 2023-11-16
> >
> > **Comparing Benchmarks With Different Classes**:
> >
> > We have a few main reasons for evaluate the full ImageNet accuracy instead of a class subset:
> > 1. To align with the robustness literature, in which the robustness benchmarks (Imagenet-R, Sketch, etc.) similarly contain a subset of ImageNet classes and the accuracies are directly compared to ImageNet.
> > 2. To provide consistent evaluations across robustness benchmarks such as ImageNet-Sketch/ImageNet-R, etc. As our goal is to evaluate these geographic benchmarks along with the robustness benchmarks, our analysis would be limited to just a very few classes if we were to select only classes that were present across all benchmarks.
> > 3. To align with existing use of DollarStreet and GeoDE in fairness, which report accuracies on the whole of each benchmark, along with ImageNet’s accuracy with all classes.
> >
> > **Additional References for Domain Adaptation**:
> >
> > Thank you for pointing these works out! We added discussion of these works in Section 2  in the updated paper.
> > Including Other Region Comparisons: We agree that other subgroup comparisons would be interesting! We chose to focus on Europe-Africa disparities in our plots, as those correspond to best and worst performing regions across models, which also aligns  with worst-group accuracy in the robustness literature. However, we included the results with Europe vs each of the other regions, and find that our results hold. There are significant progress gaps across these regions and across both datasets, with Europe-Asia having a slope difference of 0.17 / 0.12 for DollarStreet and GeoDE respectively, and Europe-Americas having a slope difference of 0.09 / 0.09 for DollarStreet and GeoDE. As expected, these gaps are smaller than the best/worst subgroup comparison of Europe/Africa, but these progress gaps are just as pervasive across other region pairs and we agree it is interesting to analyze (See Appendix G for plot).
> >
> >
> > **Comparing GeoDE and DollarStreet Results**:
> >
> > Great question - there are a few main drivers of this different from our perspective (below). We have added a discussion section of this in Appendix A.
> > 1. DollarStreet is more diverse in terms of geography and household income. DollarStreet was curated to explicitly capture a broad set of households, sending photographers to over 63 countries to people’s homes, and selecting households with a variety of incomes. GeoDE was crowdsourced among internet users from 19 countries and was designed to show the importance of crowdsourced data compared to internet scraped data.
> >
> > 2. GeoDE has a 1-to-many label mapping, whereas DollarStreet has a 1:1 label mapping. As shown in Appendix A.1 , GeoDE classes are coarser as there are GeoDE classes that map to as many as 25 ImageNet classes, while DollarStreet has 1 ImageNet label for each DollarStreet label. This makes GeoDE an easier dataset than DollarStreet for ImageNet models (we also evaluate a large number of zero-shot models that use the ground truth labels, without requiring mapping).
> >
> > Despite the large differences in dataset curation and difficulty, we find very similar results on DollarStreet and GeoDE, indicating that the problems we discover are not specific to a given dataset curation, or labeling. We find that both datasets have large geographic disparities, even with SOTA models. Most critically, we find for both benchmarks that disparities are not resolved by progress on standard generalization benchmarks, dataset scaling, standard robustness interventions, or architecture scaling.

---

> > > ### Author Response · Authors · 2023-11-22
> > >
> > > Hello reviewer PnXg, thank you again for your excellent suggestions and thorough consideration of our work! We would be grateful if you can confirm whether our response has addressed your questions, and let us know if any issues remain - we remain available for discussion. To recap our response:
> > >
> > > **Mechanisms of Geographical Disparity**: We are excited to have added a section (highlighted in blue, Appendix I) describing different reasons for geographical disparities, and believe this discussion significantly improves the clarity and impactfulness of our work. We are also excited to report that we scaled our KNN inference experiment  in Appendix F to the whole of deduplicated LAION400M, rather than a random sample. We found that **our results hold at scale, approximating that almost 60% of LAION samples are from western regions, and finding Europe has 2.24X more images than Africa**. This indicates that the empirical trends we discover in this work are likely to be data-driven, and more careful data curation with respect to geography could be a meaningful research direction identified by our findings. We hope that our work provides strong empirical evidence for the pervasiveness and importance of geographic disparities, as well as detailed insight into this challenging open problem that can guide future work.
> > >
> > > **Label Noise:** we agree that label noise is an important consideration, and we treat it carefully in our work. We help control for this with a suite of zero-shot models, which do not rely on label mappings and show consistent geographic disparities.
> > >
> > > **Differences between GeoDE and DollarStreet**: We **added a new section to our revised version** describing these dataset differences in Appendix A  (highlighted in blue in our revised submission), including an outline of the considerations we made in ensuring useful and consistent evaluations with existing literature and robustness benchmarking. Thank you for the question, we think this is an important point and improved our paper!
> > >
> > > **Domain Adaptation References:** We greatly appreciate highlighting these works from domain adaptation, and we have **added them to our related works** (highlighted in blue in our revised submission). We agree that analyzing GeoNet would be an interesting addition to our analysis with DollarStreet and GeoDE. Given the large number of models in our evaluation suite, we don't believe we will have time to replicate our analysis on this additional dataset, but it is a useful additional analysis and are working on incorporating into future work. Thank you for the excellent suggestion!
> > >
> > > **Analyzing Other Region Comparisons**: We agree this is quite interesting, and have **added figures with other region pairs in Appendix G**, finding consistent results with our best/worst region evaluation we use in our figures (Europe/Africa). Thank you for your suggestion!

---

### Official Review · Reviewer_QyGA · 2023-11-01

**Soundness:** 2 fair
**Presentation:** 3 good
**Contribution:** 2 fair
**Rating:** 5
**Confidence:** 3

**Summary:**

To study the reliability of foundation models, this paper proposes to evaluate these models on crowdsourced, global datasets with natural distribution shift. The paper provides evaluation of 100+ vision models on the benchmark datasets DollarStreet, GeoDE, and compare to the evaluation on standard benchmark datasets ImageNet. The findings show that existing evaluation on standard benchmark dataset is limited, and it is promising to use more curated and/or representative datasets for evaluating foundation models.

**Strengths:**

The studies problem of this paper is important to the community. In the era of foundation models trained on billion-scale dataset, it is important to re-consider the proper datasets and metrics for evaluation.

The paper provides interesting findings that show the importance of geographic factors in model evaluation.

**Weaknesses:**

Although the paper studies an important problem, the technical contribution is limited, and the main findings are mostly empirical.

The paper mainly consider the geographic factor in evaluation, but does not provide a more comprehensive review, discussion or comparison on the other factors. For instance, most existing CLIP, OpenCLIP models are evaluated on a diverse set of datasets for completeness. Would the dataset diversity be another important factor?

The paper does not provide a very clear and conclusive discussion to point out the potential direction/solutions for resolving the data challenges in evaluation. For instance, what factors should be considered in designing the proper benchmark datasets? What are the right evaluation metrics to consider for evaluating foundation models?

**Questions:**

Please provide more discussion on the following questions:

What are the most important factors to consider to design the proper benchmark datasets for evaluating foundation models?

What are the proper evaluation metrics to evaluate these factors?

What are the solutions/directions to design the proper evaluation metrics and benchmarks?

**Details Of Ethics Concerns:**

No concern on Ethics.

---

> ### Author Response · Authors · 2023-11-16
>
> Thank you for your appreciation of our insights as important to the community, and especially relevant in the era of foundation models trained on large-scale web data. We are glad that you found our results interesting!
>
> **Our primarily empirical analysis as limited**:
>
> We agree that our work is largely empirical. However, we disagree that empirical work has limited contributions. We believe that some of the most impactful work in machine learning has been work that provides scientific insight, and redirects the field to new problems, new model limitations, or new approaches . Our intention with this work is to highlight geographical disparities as an open and practically important scientific challenge for the research community, and we support this with **actionable insights** below
> 1. Our work shows for the first time how extensive geographic generalization failures are across model architectures, training paradigms, and research datasets.
> 2. We discover that these failures, unlike those represented in existing benchmarks and counter to researcher intuition, are unsolved by current mitigations, data scaling, or model scaling.
> 3. These results indicate that standard measures of progress have over-indexed on ImageNet and ImageNet-based metrics, leaving geographical disparities as an understudied and critical problem affecting real-world deployment.
> 4. We offer new ways to both measure and mitigate this problem, with our KNN rebalancing experiment, and last layer retraining experiments. We also release a simple-to-use benchmarking library allowing researchers to evaluate models with just 4 lines of code.
>
> **Discussion on Dataset factors and selection**:
>
> Thank you for this insightful suggestion! We absolutely want to emphasize the importance of data selection as a main takeaway of our work. We added some language to our discussion and an **additional section in Appendix H** describing these factors to our paper. To your specific questions:
> 1. **Would dataset diversity be an important factor in evaluation?** Absolutely! Our empirical evidence indicates that geographical diversity is a missing axis from standard benchmarks, and we hope to encourage the field to consider geographical disparities (and all data subgroups) as important factors for evaluation.
> 2. **What factors should be considered in designing proper evaluation benchmarks?** Our work provides evidence that data sources, label curation, geographic representation, and income information should all be closely considered. In Appendix H, we emphasize that the best evaluation metrics are crowdsourced and grounded in real-world use, which is a significant difference between standard benchmarks and our geographic benchmarks.
> 3. **What are the right evaluation metrics to consider for evaluation foundation models?** We address this question by highlighting the need for using subgroup evaluation metrics in generalization literature, and the limitations of using average accuracy alone. We will cite relevant examples of subgroup analysis in fairness and spurious correlations. Similarly, our work highlights the impact of looking more closely at the kinds of accuracy gains made (improvement on which portions of data), which we demonstrate with our progress gap measure. Overall, we hope to suggest that there is no one particular benchmark or tool that can perfectly measure model reliability, but that having a comprehensive suite of measures, and by analyzing benchmarks more precisely, we can improve our understanding of model behavior and its reliability.
>
> Thank you for your insightful suggestions! We welcome any other questions or suggestions to improve the clarity and impact of our work.

---

> > ### Author Response · Authors · 2023-11-22
> >
> > Hello reviewer QyGA, thank you again for your excellent suggestions! We would be grateful if you can confirm whether our response has addressed your questions, and let us know if any issues remain - we remain available for discussion. To recap our response:
> >
> > **Our work is primarily empirical, but has broad and impactful contributions**: We summarize the novel insights uncovered by our work, and the significant implications for the field. We discover an alarming trend, finding geographic disparity as a pervasive problem unsolved by standard benchmark progress, scaling, or common interventions, and worsening over time. We hope that our work demonstrates the impact of this problem, and provides both motivation and insight for new directions of research in this area, including in careful data rebalancing and in design of specific geographic mitigations.
> >
> > **Emphasizing the Importance of Dataset Factors and Benchmarks**: We found these suggestions to greatly improve the impact of our insights, and have added two sections in the Appendix thoroughly discussing these points, highlighted in blue in the revised submission. We hope that these new sections have expand the impact of our insights by providing a framework to discuss data factors that drive these reliability failures. Thank you for the excellent suggestion!

---

### Official Review · Reviewer_hbmo · 2023-11-04

**Soundness:** 3 good
**Presentation:** 3 good
**Contribution:** 3 good
**Rating:** 6
**Confidence:** 5

**Summary:**

In this work, authors target an important aspect of model training: the data imbalance across region. Author used  two globally crowdsourced datasets (DollarStreet and GeoDE), as calibrated geographic disparity measurement, and shows that model train on conventional dataset, like imageNet, are highly dominant by the west. And model perform better on conventional dataset will enlarge this geographic disparity across regions. Lastly, author propose to solve this problem by last layer fine-tuning on geographic balanced dataset.

**Strengths:**

1. The problem author targeting, is of significance to the community, especially in the foundation model, for which data are more dominant than the model itself.
2. Author had made significant empirically contribution by experiment on a large number of models.
3. Author have identified data imbalance in standard benchmark, and shows that improvement over conventional evaluation will exacerbate geographic disparities.
4. Author shows that applying conventional trick like data augmentation and scaling won’t solve this problem.
5. Author also propose a simple fix of adopting last layer fine-tuning over geographic balanced dataset.

**Weaknesses:**

Although the concept of adopting geographic disparities is a neat measurement for data bias, author limit the measurement of such concept only on two specific dataset, which make the final ‘improvement’ less convincing. Also, the proposed solution to resolve this bias by last layer fine-tune, is a common approach in data debiasing and less novel, especially author tried to fine-tune on a geographic dataset.

One potential significant improvement of this work, in my opinion, is to apply author’s insight onto standard dataset. For instance, in appendix F LAION CLUSTERING EXPERIMENTS, author use two geographic dataset only as cluster center to measure the ‘geographic disparities’ for LAION datasets. It would be much stronger an insight if authors shows that finetuning/retraining model with ‘balanced LAION’ dataset could reduce geographic disparities.

**Questions:**

Plase refer to weakness.

---

> ### Author Response · Authors · 2023-11-16
>
> Thank you for your thoughtful appreciation of our work! We are glad that we were able to convey the impact of our empirical contributions, the novel insights of the geographical dataset imbalances we characterize, and the crucial implications of our results for benchmarking.
>
> **Using 2 Benchmarks for Evaluation of Geographic Disparities**:
>
> Our analysis does rely on two benchmarks to characterize geographical disparity, and we agree that our results (as with all research analyses), should be interpreted considering the datasets used. However, we emphasize that these benchmarks are reliable measures of geographical disparities, as they are 2-6x larger than existing robustness benchmarks, and are supported by a wealth of research use in fairness [1,2,3]. DollarStreet and GeoDE have 61K and 38K respectively, compared to 10K in ImageNet-A and 30K samples in ImageNet Sketch. We have added a paragraph in the Appendix A (in blue) to clarify our use of these benchmarks, and welcome any further suggestions.
>
> [1] Shankar et al, No Classification without Representation: Assessing Geodiversity Issues in Open Data Sets for the Developing World
> [2] Gustafson et al, Pinpointing Why Object Recognition Performance Degrades Across Income Levels and Geographies
> [3] Xiao et al, Noise or Signal: The Role of Image Backgrounds in Object Recognition
>
>
> **Balanced Retraining on LAION**:
>
> Thank you for this insightful suggestion! In our work, we demonstrate how geographical disparity has worsened with progress on standard benchmarks, is  exacerbated by data and model scaling, and produce some of our preliminary evidence of dataset imbalance in LAION as a contributing cause. We agree that the next step in this research direction is to show how our insights can be used to design rebalancing mitigation strategies for large-scale training datasets. We find this suggestion of performing a rebalancing from our LAION cluster analysis, and are excited to try it. This is quite a computationally demanding experiment, which requires balancing a very large scale dataset, coordinating distributed training, etc. which is why we did not explore a rebalancing mitigation in the original submission. However, as we wrote in Appendix F, our clustering analysis should be viewed as a rough proxy of the regional distribution, as there are significant limitations - most notably, that LAION has several orders of magnitudes more concepts represented than DollarStreet. Therefore, rebalancing LAION based on comparisons to DollarStreet may not balance the dataset well across concepts.
> We are excited to try this however, and will share results when we have them.
>
>
> **Contribution of Last Layer Retraining**:
>
> We agree that last layer retraining has certainly been used in other works, which we cite in Sec. 6.3. In this work, we demonstrate empirically that geographical disparities are a pervasive problem, and that existing methods, such as scaling and standard robustness interventions, fail to consistently improve it (Figure 3 and Table 3). Our intention with our experiment in 6.3 is not to claim the development of a new method itself, but to show how, counter to these other approaches, last layer retraining could be a practical mitigation for improving geographical robustness. We expand on existing work by showing how last layer retraining is not just allowing adaptation to a given geographic distribution ([1]), but that last layer retraining improves geographic robustness on new datasets (without requiring the monumental task of generating labels for web-scale data). We have added some (highlighted) language in Sec. 6.3 to clarify.
>
> We welcome any other questions or suggestions to improve the clarity and impact of our work!
>
> 1. Vikram V Ramaswamy, Sing Yu Lin, Dora Zhao, Aaron B Adcock, Laurens van der Maaten, Deepti Ghadiyaram, and Olga Russakovsky. Beyond web-scraping: Crowd-sourcing a geographically diverse image dataset. arXiv preprint arXiv:2301.02560, 2023.12

---

> > ### Author Response · Authors · 2023-11-22
> >
> > Hello reviewer hbmo, thank you again for your insightful questions! We would be grateful if you can confirm whether our response has addressed your concerns, and let us know if any issues remain. To recap our response, we make the following points:
> >
> > **Using 2 Benchmarks for Geographical Diversity**: We show how our 2 benchmarks are robust measurements of geographical disparities, due to them being much larger than standard robustness benchmarks, and whose use are supported by longtime use in fairness.
> >
> > **Balanced Retraining on LAION**: We are excited to report that we have made significant progress in adding to our experiment in Appendix F, and have **scaled our KNN inference experiment to the whole of deduplicated LAION400M**, rather than a random sample. We found that our results hold at scale, approximating almost 60% of LAION samples are from western regions, and finding **Europe has 2.24X more images than Africa**. This indicates that the empirical trends we discover in this work are likely to be data-driven, and more careful data curation with respect to geography could be a meaningful research direction identified by our findings. We are working on a finetuning of CLIP based on these findings to further emphasize the directions produced by our insights.
> >
> > **Last layer retraining:** We clarify that the contributions of our last layer retraining experiment are not to introduce a novel method, but to show how last layer retraining could be an effective mitigation for improving geographical disparity. We believe the particularly compelling part of that experiment is how geographical robustness was improved not only on the retraining dataset (DollarStreet), but also a new dataset with a different distribution (GeoDE). We hope we have clarified these contributions in the highlighted language in Sec 6.3.

---

> ### Comment · Reviewer_hbmo · 2023-11-23
>
> Thanks author for the detail reply and providing additional experiment. I am satisfied with where this paper is heading toward thus change my rating to 6 and soundness to 3.
> In your future revision, please make sure to include result of retraining CLIP on balanced LAION dataset, and publicly release the label for geographical property of each LAION sample. Also you should include a link in the main text(where you mentioned to your comment on 'Benchmark Use' in appendix.

---

### Author Response · Authors · 2023-11-22

We thank the reviewers for their careful reading of our work and insightful questions! We believe their feedback greatly improved the quality of our paper.

All reviewers agreed that our work addressed a **problem of critical importance** (“very pertinent problem” [PnXg], “important to the community” [QyGa], “significance to the community” [hbmo], and “important topic” [NCcF] ). They also praised our **extensive analysis**  (“extensive empirical evaluation” [PnXg], “large scale evaluation”[NCcF], “significant empirical contribution” [hbmo]). We were glad to see that reviewers appreciated our work’s insights (“important observations” [PnXg], “Interesting findings” [QyGA, NCcF]). We were particularly pleased that they **commended our work as having important implications for the field** (“...reconsider proper datasets and metrics for evaluation” [QyGA], “...spur future directions” [PnXg], “[findings] are worth emphasizing [NCcF]”, “identify imbalance in standard benchmark” [hbmo]). Finally, we are glad that reviewers found our work to be “well written and easy to understand” [NCcF], with a “balanced perspective” [QyGA] on an open challenge [“PnXg”].

Reviewers asked for more discussion of the underlying mechanisms for our empirical results. We are excited to share that we have **added a paper section in Appendix I** outlining perspectives on this open problem, and **also expanded our KNN analysis in Appendix F to approximate the regional distribution of LAION400M**. This analysis approximates that 60% of LAION400M is from western regions, having 2.24X more data from Europe than Africa. This analysis provides compelling additional evidence for the role of training data diversity in the concerning disparity trends we uncover in our work. We hope that this additional discussion section and the additional experimental results empower future work on the important challenge of geographic disparities.

Reviewers also asked about the implications of work in dataset design and evaluation, and we are **thrilled to include an additional section in Appendix H** discussing the impacts of our findings in these areas. We again hope that this discussion will provide a framework for future research in this open problem.

Reviewers also asked for some additional analyses, including plots with other region subgroup pairs, and analysis of geographic disparities on DollarStreet while controlling for household income. **We were happy to perform these analyses, and our findings from these were consistent with the original analysis** (Appendix G and Appendix C).

Finally, reviewers asked for more in-depth discussion of our benchmarks, and for clarification for our interpretation of results in Sec. 6.3. **We added language to Appendix A, related works, Sec. 6.3, and discussion to improve the clarity of our writing.**

We hope reviewers take these substantial revisions and multiple additional experiments based on their feedback in their final assessment of our work. We remain available for discussion and to answer any outstanding questions.

---

### Meta-Review · Area_Chair_zokZ · 2023-12-05

**Metareview:**

The paper studies how progress on ImageNet (which is not quite all of object recognition) compares to progress on geographically more diverse data (DollarStreet, GeoDE).
The strengths of the paper (as noted by the reviewers):
+ Paper studies an important problem
+ The paper highlights the importance of geographic factors
+ The rebuttal did provide some larger scale experiments (on LAION)
The authors were able to answer all the questions of the reviewers.
In the end, three reviewers recommend acceptance, one borderline rejection. The AC agrees that there is sufficient substance here to warrant acceptance.

**Justification For Why Not Higher Score:**

The paper has hints of exciting results (e.g. the LAION results in the rebuttal), but falls short of a truly convincing experimental validation (ultimately recognition != ImageNet, and most of the datasets evaluated later are derivatives of ImageNet).

**Justification For Why Not Lower Score:**

Nobody really wants to reject the paper (me included). Reviewer QyGA (borderline reject) didn't update his review after the rebuttal.

---

### Decision · Program_Chairs · 2024-01-16

Accept (poster)